# Color Conversion Light-Emitting Diodes Based on Carbon Dots: A Review

**DOI:** 10.3390/ma15155450

**Published:** 2022-08-08

**Authors:** Danilo Trapani, Roberto Macaluso, Isodiana Crupi, Mauro Mosca

**Affiliations:** Thin-Films Laboratory, Department of Engineering, University of Palermo, Viale delle Scienze, Bdg. 9, I-90129 Palermo, Italy

**Keywords:** carbon dots, color conversion, white light-emitting diodes, multicolor light-emitting diodes, carbon-dot-based light-emitting diodes, phosphors, organic materials, LEDs

## Abstract

This paper reviews the state-of-the-art technologies, characterizations, materials (precursors and encapsulants), and challenges concerning multicolor and white light-emitting diodes (LEDs) based on carbon dots (CDs) as color converters. Herein, CDs are exploited to achieve emission in LEDs at wavelengths longer than the pump wavelength. White LEDs are typically obtained by pumping broad band visible-emitting CDs by an UV LED, or yellow–green-emitting CDs by a blue LED. The most important methods used to produce CDs, top-down and bottom-up, are described in detail, together with the process that allows one to embed the synthetized CDs on the surface of the pumping LEDs. Experimental results show that CDs are very promising ecofriendly candidates with the potential to replace phosphors in traditional color conversion LEDs. The future for these devices is bright, but several goals must still be achieved to reach full maturity.

## 1. Introduction

Luminescent carbon dots (CDs) are a new form of nanocarbon quantum dot (QD) that have gained a huge amount of interest in recent years for their properties; in particular, their optical properties make them suitable for light-emitting diode (LED) manufacturing. When, in 2004, the group of W. A. Scrivens of the University of South Carolina (US) [1] accidentally discovered CDs, they could not imagine the success of their breakthrough. CDs are very versatile zero-dimensional nanomaterials of about 10 nm diameter or less [2], that are based on carbon. To date, they are proven to have a number of remarkable properties, such as low toxicity and high biocompatibility [3], extremely easy production processes, abundance of precursors [4], easily tunable surface proprieties [5], and very bright and stable photoluminescence [6], comparable with those of conventional quantum dots [3].

CDs are studied for their enormous range of possible applications, such as photovoltaics [7], drug delivery [8], bioimaging [9], sensing [10], color conversion [6], and so on [11]. Most of the research conducted so far is focused on the surface properties, because they can be easily modified by changing the amount and the type of superficial bonded chemical species. Their optical properties make CDs very promising as the color conversion layer for high color rendering index (CRI) multicolor light-emitting diodes (LEDs), white LEDs (WLED) [12], and even TV displays [13].

In recent years, the number of articles regarding CDs has exponentially increased since the first paper published in 2004, concerning their synthesis [1]. Nowadays, thousands of highly cited papers have been published, as shown in Figure 1.

The synthesis of CDs is one of the most important steps for the determination of optical and physical properties, because CDs can be functionalized through specific superficial groups and species. Essentially, there are two major categories of production methods: bottom-up and top-down processes. The difference between these two approaches lies in the kind of precursor materials used. The bottom-up approach is based on the carbonization and polymerization of small molecules of carbon precursors into CDs of different size and properties depending on the chemical process used. Most common methods are microwave-assisted pyrolysis [14], and solvothermal or hydrothermal treatment [15]. The top-down approach, instead, refers to the fragmentation of macromolecules of carbon precursors into smaller size particles by different methods, such as laser ablation [16], acidic oxidation [17], and arc discharge [1].

Typically, all these methods produce CDs dispersed in liquids, such as water or organic solvents. The type of solvent determines some of the photoluminescence properties, such as the quantum yield (QY), and causes a shift in the emission spectrum [18]. The importance of a liquid solvent lies in the ability to disperse the CDs within it. When the concentration of CDs is too high, or when they are present in solid-state form, photoluminescence quenching occurs. This problem, caused by the aggregation of CDs [19], has hindered the application of CDs in the development of optoelectronic devices [20].

This review has the intent to summarize and describe the state-of-the-art techniques and the progress in the development of color conversion multicolor LEDs and WLEDs based on CDs. These devices are based on the effect of frequency down-conversion (or color conversion) occurring when a phosphor layer, typically emitting in the visible spectrum region, is irradiated by a primary source, such as a blue, violet, or near-UV LED. If irradiated with a high frequency source, CDs emit in the visible region (at lower frequency than that of the source), and can be exploited to fabricate multicolor and white light-emitting devices.

## 2. Materials for Frequency Down-Conversion LEDs

It is public knowledge that Nick Holonyak Jr. discovered, in 1962, the light emitting capabilities of a diode in the visible spectrum [21]. However, around the time of their discovery, LEDs were a very low efficiency light source, used for signalizing or as indicators in laboratory electronic test equipment until 1976, when Thomas P. Pearsall designed the first high-brightness and high-efficiency LED for optical fiber telecommunications [22]. To date, LEDs of every size and emission wavelength have been readily available on the market, even for lighting purposes and image display.

One of the most frequently used methods for obtaining different emission spectra is color conversion, typically used for high CRI and white LEDs and displays, where the lowest emission wavelength source pumps other layers of materials, commonly called phosphors, that induce a conversion to the highest wavelengths.

### 2.1. Phosphors

Briefly, phosphors are particular materials or compounds that present an energy gap between the valence and the conduction band, in a similar way to semiconductors. To achieve light emission, the transition between these two bands must be “direct” or without any change in the electron momentum. The color conversion ability of these materials lies in the different energy levels existing between the valence and the conduction band. These levels, or traps, are generated by lattice defects or impurities present in the crystal lattice, and introduce non-radiative paths that dissipate energy without the emission of photons. The color conversion process is due to the distribution of these trap levels and the energy gap of the material: when a phosphor is exposed to a photon with higher energy than its energy gap, an electron from the valence band can absorb this energy and be excited to a higher energy level. This level could be the edge of the conduction band or a trap level. Since trap levels typically dissipate energy in a non-radiative way, transitions between them cause a decrease in electron energy, and when the direct transition occurs the relative photon will be emitted with the remaining energy, therefore with a higher wavelength than the absorbed photon. This process, illustrated in Figure 2, is known as phosphor-based frequency down-conversion, and represents the base of color conversion.

Usually, phosphors consist of an inorganic host material, such as transition metal compounds, where defects in the crystal lattice are introduced by dopants, dislocations, or impurities, which are called activators. The most frequently used materials for phosphors are sulfides, selenides, and cadmium, and different rare earth materials, such as cerium, with added activators. Commercial white LEDs are typically attained by an InGaN/GaN blue LEDs pumping Ce:YAG phosphor powder dispersed in a polymeric matrix material. These materials could cause different problems in the foreseeable future because of their toxicity, low biocompatibility and, in some cases, availability difficulties. In other words, they may be non-sustainable materials.

### 2.2. Quantum Dots (QDs)

Another way to make color conversion possible is the use of quantum confinement effects. Quantum confinement occurs when the charge carriers of a material are confined in a space region smaller than its Bohr exciton radius [24]. In other words, looking at the charge carriers as waves, the region of space must be smaller than the de Broglie wavelength of the electron [25], which can be quite different from material to material, and is determined by the quantum mechanical nature of the electrons and holes in the materials [26]. In such cases, electrical and optical properties of the material change, leading to the formation of discrete energy levels instead of bands, and their position and energy gaps depend on the dimension of the confinement region [27]. The quantum confinement could be attempted along the three dimensions, giving rise to three types of confinement structures: mono-dimensional confinement or quantum well, bi-dimensional confinement or quantum wire, and three-dimensional confinement or quantum box/dot [26]. These structures are typically called 2D, 1D, and 0D, respectively, according to the remaining degrees of freedom. As can be seen in Figure 3, all these structures exhibit different densities of state compared to bulk material and to each other [26].

Color conversion in QDs acts in a similar way of phosphor, but in this case the different levels and gaps are given by the quantum confinement effects instead of lattice defects or impurities, and could be tuned simply by varying both the dot dimensions and the materials used to synthesize QDs. The bandgap energy, which determines the color of the fluorescent light, is inversely proportional to the square of the size of the quantum dot [29,30]. Larger QDs have more energy levels that are more closely spaced, allowing the QD to emit (or absorb) photons of lower energy. In other words, the emitted photon energy increases as the dot size decreases, because greater energy is required to confine the semiconductor excitation to a smaller volume [31]. Although QDs seem to be the future of illumination, display, and optoelectronics, the typical materials used to create QDs are binary compounds, such as lead sulfide [32], lead selenide [33], cadmium selenide [34], cadmium sulfide [35], cadmium telluride [36], indium arsenide [37], and indium phosphide [38]. Dots may also be made from ternary compounds, such as cadmium selenide sulfide [39]. Furthermore, recent advances have been made concerning the synthesis of colloidal perovskite quantum dots [40]. Although QDs are very popular due to their promising optical performances, they are problematic because of their content of heavy metals and toxic materials, which poses a serious health risk to most living beings due to cytotoxicity [41]. Additionally, heavy metals are well-known environmental pollutants due to their toxicity, persistence in the environment, and bioaccumulative nature [42]. In recent years, research has focused on different and less troublesome materials, and much interest has been aroused by CDs, which are non-hazardous and biocompatible QDs based on carbon.

### 2.3. Carbon Dots (CDs)

CDs are very promising zero-dimensional structures because they are based on carbon, which makes them environmentally friendly. Additionally, they exhibit low toxicity, good water solubility, and chemical stability [43]. CDs also present very good optical properties, such as strong photoluminescence, optical tunability, luminous stability, and a very high photoluminescence QY [5,44,45,46], comparable with those of traditional inorganic QDs. Today, a photoluminescence QY of over 80% has been reached [47,48].

CDs consist of two parts: one is a spherical-like core, formed by the stacking of multiple graphene fragments in an ordered or disordered manner; the other is rich in functional groups distributed on the surface of CDs [43]. In general, the fluorescence behavior of CDs is controlled by the relationship between the carbon core and surrounding chemical groups [49]. The color of the fluorescence is determined by several features, such as the electronic bandgap transitions of conjugated π-domains, surface defect states, local fluorophores, and the doping element present in the dot structure [50]. Moreover, the light emission intensity depends on different mechanisms, mainly π-domains, surface states, and molecule states [50]. For CDs with large conjugated π-domains and few surface chemical groups, the light emission is mostly due to conjugated π-domains, which are considered the carbon core state fluorescence centers. The bandgap of conjugated electrons derives from quantum confinement effects; thus, the emission color of CDs can be adjusted by tuning the size of the conjugated π-domains [51].

Surface defects contribute to light emission mechanisms, the surface chemical groups of CDs having various energy levels that cause different emission wavelengths by radiation relaxation. Furthermore, sp^3^ and sp^2^ hybrid carbon on the surface of CDs, and other surface defects, can lead to multicolor emissions from their local electronic states [50]. When CDs are excited by light, electrons accumulate in adjacent surface defect traps, and return to the ground state emitting visible light at different wavelengths. In this case, the emission color depends on the number of surface defects. Increasing the degree of surface oxidation also increases the number of surface defects, leading to a red shift in emission wavelengths [50]. Another contribution to light emission derives from small organic molecules that through the carbonization process become fluorophores. These could be attached to the surface or the interior of the carbon skeleton, and emit light independently [50]. In general, the QY of fluorescent molecules is higher than the QY of the core fluorescence, but the molecular luminescence stability is lower.

## 3. CD Photoluminescence Quenching

CDs also have several drawbacks, as they suffer from photoluminescence quenching in the solid state. This problem is related to a lot of different mechanisms (some of which are not well understood at present), due to the proximity and even collisions between CDs or between CDs and other impurity particles [19]. In recent years, this problem has largely been analyzed in order to fully exploit the CD luminescence properties.

From an optoelectronic point of view, quenching is a major problem that limits the use of these materials for lighting and similar purposes. Quenching mechanisms of CDs are related to the interactions between CD particles and some other impurities, and the way these interactions occur determines the type of quenching effect. The best-known quenching mechanisms include: (i) static and dynamic quenching, (ii) inner filter effect (IFE), (iii) photoinduced electron transfer (PET), and (iv) energy transfer [19]. The latter is also divided into Förster resonance energy transfer (FRET), Dexter energy transfer (DET), and surface energy transfer (SET).

(i) Static quenching involves the formation of a complex by the combination of ground-state CD molecules with each other or with other molecules, named quenchers [52]. When the complex absorbs light, it immediately returns to ground state without the emission of photons. Dynamic quenching is caused by the collisions of excited CD molecules with quenchers [52], and the excited state returns to ground state before photon emission occurs. (ii) The second quenching mechanism (IFE) can be called “apparent quenching” [19] because it is not a real quenching process, but a reduction in the emission or absorption rate due to the reabsorption of CD samples. There is a primary IFE that is caused by the absorption of excitation light as it travels through the sample, leading to non-uniform sample illumination [53]. Only the external part manifests the relevant emission rate. The secondary IFE, instead, is caused by reabsorption of emitted light as it propagates through the sample, causing a red shift in the emission spectrum [53]. (iii) PET occurs when excitation by light absorption changes redox properties of the material, causing non-radiative energy dissipation by electron transfer from a donor to an acceptor in the ground state; different from DET, a radical ion pair is formed and the system relaxes to the ground state via charge recombination [54]. Note that PET may lead not only to quenching, but also to a broadening of the emitted luminescence. This was observed, for instance, in several benzoxazole and benzothiazole compounds [55]. While in dilute solutions, only small energy differences are reported between excitation and fluorescence O–O bands, instead in solid state, these compounds lacking in side groups exhibit a marked broadening of the emission luminescence spectrum. This molecular configuration allows benzoxazole and benzothiazole compounds to experience PET when in a solid state. A detailed discussion on this issue is reported in [55]. (iv) Finally, we consider the energy transfer mechanisms. FRET occurs when the CD emission spectrum overlaps the quencher absorption spectrum. In this case, if an appropriate distance (Förster distance, R_0_) is present between the CDs and the quencher, the energy transfers from the CD’s excited states to the quencher’s ground state occurs through dipole–dipole interactions, without the emission of photons. In DET, the transfer occurs via electrons that move from one molecule to another along a non-radiative path, during which there is a wavefunction overlap between the two molecules [56]. SET is very similar to FRET in terms of the dipole–dipole interactions, but for SET, these occur at nanoparticle surfaces with an isotropic distribution of dipole vectors, and at a much slower decay rate [57].

## 4. CD Preparation

CDs are prepared via various methods. Carbon precursors are very abundant and disparate, and the variation in precursors can lead to completely different behaviors and performances. As mentioned earlier, there are two distinct families of production methods, bottom-up and top-down. These two terms are used in a wide variety of fields, and indicate the kind of approach with which a problem is addressed. The bottom-up approach is used to create a system (up) starting from the constituent single pieces (bottom). The top-down approach, instead, is used to solve the problem starting from an overview of the entire system (top) and then breaking it down to make refinements. In the case of CDs, these two approaches refer to the type of precursor used and how they are managed. A bottom-up synthesis starts from small molecules of carbon precursors, such as urea [20], citric acid [58], sodium citrate [59], glucosamine [60], ascorbic acid [61], ethanol [62], and so on, that are typically dissolved in water or organic solvents. The formation of CDs starts when the precursors are mixed, and the carbonization and polymerization is catalyzed by a thermal process, such as microwave-assisted pyrolysis [14], or solvothermal or hydrothermal treatment [15].

Among the bottom-up methods is the particularly unusual gaseous detonation approach proposed by He et al. [63]. These authors used citric acid and urea as the precursors, which were inserted into a detonation tube, which, in turn, was vacuumized and heated to 85 °C. After hydrogen and oxygen were pumped inside the detonator, the mixture was detonated. Colloid products were produced and, after freeze-drying, CD powder was collected. This method is probably the fastest way to prepare CDs showing solid-state fluorescence (SSF).

Top-down synthesis involves the breaking down of large carbon aggregates and structures into smaller molecules or CDs. Typical precursor structures used are carbon nanotubes [64], activated carbon [65], graphite [66], carbon soot [17], with methods such as laser ablation [16], acidic oxidation [17], arc discharge [1], electrochemical synthesis [64], etc. The first attempts to create CDs were made using a top-down approach. Xu et al. in 2004 synthesized the first CDs during the purification of single-walled carbon nanotubes (SWCNTs) from arc discharge soot [1]. This method produces very impure SWCNTs, therefore purification is needed. They performed electrophoretic purification by oxidation of arc discharge soot with HNO_3_, and then extraction with basic water. This process formed a suspension of SWCNTs with other soluble impurities, which were then subjected to gel electrophoresis through agarose gel slabs. Three kinds of materials were isolated: long nanotubes, short irregular tubular material, and highly fluorescent particles of about 1 nm diameter, later named CDs. In 2006, Sun et al. [66] used laser ablation of a carbon target in the presence of water vapor with argon as the carrier gas. The carbon target was prepared by hot pressing a mixture of graphite powder. The obtained CDs were available in aggregate form, and showed photoluminescent properties only after surface passivation with other organic species, such as polyethylene glycol (PEG).

Although many (even recent [67]) papers report top-down synthesis of CDs, this kind of approach is difficult to carry out, and requires sophisticated equipment. The recent trend in CD synthesis is to produce CDs in a green, economic, and simple way; therefore, the bottom-up approach is preferable, as reported in the most recent literature.

## 5. Fluorescent CDs in Solid-State Form

For years, CDs were prepared in liquid form, and dissolved in solvent because of the problem of photoluminescence quenching. So far, this problem has been hard to overcome, because CD concentration should be low to avoid quenching, but not too low to generate an appreciable level of photoluminescence. This section reports on the different ways to synthesize solid-state fluorescent CDs for LED applications.

### 5.1. Solid-State Fluorescence in Matrices

The first approach towards achieving SSF employs different kinds of matrices to embed CDs, similar to solid solvents. The fluorescent particles are dispersed, avoiding aggregation, and are therefore self-quenching. For this purpose, starch is a good candidate. Starch particles contain large amounts of surface hydroxyl groups that can absorb CDs through hydrogen bonding [68]. The resulting starch matrix is transparent to visible radiation, and a QY of 50% has been achieved with starch/CDs phosphors [68]. Javanbakht and Namazi [69] fabricated CDs from citric acid by thermal treatment and neutralization by NaOH solution. Then, they mixed native corn starch and glycerol with different molar fractions of CDs. The CDs showed high dispersibility in the starch matrix, even at 50 wt%, at which the maximum photoluminescence intensity was registered. The increase in concentration of CDs led to a red shift in the emission spectrum because of the formation of slack agglomerates of sizes under 100 nm and at concentrations up to 28.5 wt%. Similar characteristics are exhibited by the mesoporous alumina (MA), leading to excellent photoluminescence properties, and thermal and color stabilities [70]. He et al. [71] dispersed CDs into an MA matrix constructed by adding aluminum isopropoxide into a solution of Pluronic P123, anhydrous citric acid, hydrochloric acid, glacial acetic acid, and ethanol, obtaining a white solid-state powder after drying. QY increased from 33% to 46.69%, and the emission was red-shifted, with an increased concentration of CDs. Mosconi et al. [72] also reported some experiments with polymer matrices, such as poly(methyl methacrylate) (PMMA), and polyester. They fabricated CDs from chlorohydrated arginine and ethylenediamine (EDA) in ultrapure water by microwave hydrothermal treatment. QY of 18% was recorded in water solution. Other types of matrices permit one to reduce the photoluminescence quenching of CDs, or to remove it. Here, we present a non-exhaustive list of matrices effectively used to disperse CDs: nylon matrix, synthesized from hexamethylenediamine, ultrapure water, and Sebacoyl chloride solution, where CDs are dissolved directly in aqueous solution; polyester–amide matrix, based on citric acid, where pentaerythritol polymerized by microwave irradiation; polyurea–urethane matrix, obtained from castor oil heated in a nitrogen atmosphere, with hexamethylenediisocyanate and dibutyl-tin-dilaurate, where CDs are in a solution of tetrahydrofuran (THF) [72]. PMMA was also investigated by Gong et al. [73], who used it to incorporate CDs for the fabrication of solar concentrators. CDs doped with nitrogen, prepared by a one-step microwave method, were also used to fabricate solar concentrators [74]. PMMA nanocomposite slabs were prepared by bulk polymerization of MMA in the presence of a thermal radical initiator. MMA monomers and PMMA polymers were mixed and heated at 60 °C, then, Azobis(2-methylpropionitrile) was added as a thermal radical initiator. CDs were dissolved in ethanol, and the mixture was transferred into a casting mold, creating slabs that were cured at 50 °C for 16 h. As the concentration of N-CDs increased, the color of the CDs/PMMA slabs became darker. The best solution was obtained with 0.08 wt% of CDs, at which an optical efficiency of 12.23% was registered. Increasing the concentration of CDs to 0.1 wt% led to a macroscopic aggregation. The maximum emission intensity was registered at 370 nm of excitation wavelength. The CDs showed excitation-dependent photoluminescence, and the intensity of the emission spectra increased with the increase in CD concentration, and with a slight red shift in the emission peak. The red shift in the fluorescence emission peak can be attributed to the self-absorption effect of CDs due to some overlap between absorption and emission spectra.

A polyester matrix was also investigated by He and his team [75]. Specifically, they studied polyethylene terephthalate (PET) thermoplastic, searching for a way to increase its toughness by incorporating PEG into the polymer backbone structure. Interestingly, they discovered that CDs can be obtained by the direct thermal decomposition of PEG in nitrogen atmosphere, and found that a highly transparent matrix can be synthesized by incorporating PEG segments into the backbone of PET, where CDs can be formulated in situ and dispersed. They synthesized PET–PEG copolymers via step polymerization of terephthalic acid, ethylene glycol, isophthalic acid, and PEG, mixing the first three in a nitrogen-protected reactor at 230 °C and 2.7 bars, and then adding PEG and Sb_2_O_3_ as catalysts. After stirring for 3.5 h at 280 °C, the melt was cooled in water. They observed a blue intense emission under UV excitation, indicating that CDs were obtained through the thermal decomposition of PEG in N_2_ atmosphere. They registered a broad absorption spectrum with a peak at 245 nm, and an excitation-dependent emission with a maximum at approximately 460 nm, attributed to the existence of different sizes of CDs. Additionally, a red shift was observed as increasing the reaction time. When the solution was polymerized, another red shift occurred, and the emission peak shifted to 500 nm, which could be attributed to the aggregation of CDs.

The group of Wang [76] used crystalline Mn-containing open-framework matrices to embed CDs. The geometry of Mn atoms in the matrix also determines the emission spectrum due to CD–matrix energy transfer. They prepared CDs with different types of amines, from the reaction gel containing MnO–Al_2_O_3_–P_2_O_5_–H_2_O (MnAPO)-amine under the same hydrothermal conditions (180 °C, 72 h). Aluminum isopropoxide and MnCl_2_-4H_2_O were mixed in deionized water. Then, H_3_PO_4_ was added to the solution and, under continuous stirring, the amine was added. The gel was subsequently stirred for 2 h and then crystallized at 180 °C for 96 h. Therefore, in this way, CDs were fabricated directly into the matrix. The best performances were registered with CDs@MnAPO-CJ50 and Cds@MnAPO-tren, fabricated in the same way but with different amine groups: N-ethyldiethanolamine (EDEA) and triethylenetetramine (TETA), respectively.

### 5.2. Self-Quenching-Resistant CDs

The use of a medium to avoid aggregation-induced photoluminescence quenching (AIQ) is not the best solution because it needs to be not only suitable for the dispersion of CD nanoparticles, but also completely transparent to both excitation and emission radiations of CDs. One of the first attempts to produce quenching-resistant SSF CDs was carried out by Chen et al. [77]. They synthesized nitrogen-doped CDs by one-pot solvothermal treatment of poly(vinyl alcohol) (PVA) and EDA, then freeze-dried them to obtain a powder. PVA chains maintain the CD’s cores at a suitable distance from each other (within Förster distance), and this structure makes SSF possible, with strong yellow–green emission under 340 nm excitation. Instead, in aqueous solution, it exhibits a 414 nm emission [77]. The dual fluorescence behavior is caused by the aggregation in solid state, so the self-quenching in this case can be considered an advantage [77], and can be used in WLEDs with good photostability, as reported in Figure 4.

This semi-aggregated state that makes SSF possible opened a line of research for tunable fluorescence properties of CDs in solid state. Essentially the emission spectrum is heavily influenced by the aggregate particle’s dimensions and concentration-dependent properties.

Another attempt was carried out by Shen et al. [78], who used a trisodium citrate crystal matrix to incorporate CDs suppressing AIQ. This method is in the middle between matrix dispersion and self-quenching-resistant CDs because the material used for the matrix is also a precursor source for the fabrication of CDs. The reaction started from trisodium citrate (carbon source), urea (nitrogen source), and three different types of organic solvents: dimethylformamide (DMF), dimethylacetamide (DMAC), and diethylformamide (DEF). After the formation of CDs under solvothermal conditions (160 °C for 4 h), the excess trisodium citrate begins to crystallize around the formed CDs, embedding them in a matrix that constrains them and avoids contact with other CDs. The solid-state CD powder formed by this process precipitates, forming SSF powder. The emission color strongly depends on the solvent used. A tunable SSF from green to yellow is achieved by adjusting the reaction solvents. Values of QY as high as 21.6% (with DMF) are obtained, close to the values obtained in water solution, deducing that CDs are well dispersed in the matrix. Additionally, good photostability is reported, the emission intensity remaining the same after 2 h of UV irradiation at 365 nm. Finally, note that changing trisodium citrate with citric acid leads to the self-quenching phenomenon (see Figure 5).

Song et al. [79] reported a concentration-dependent fluorescence of CDs in solid state. They used waste expanded polystyrene (WEPS) in dichloromethane (DCE) for solvothermal treatment, adding a microvolume of HNO_3_ into the solution, and heating for 5 h at 200 °C. As the volume of HNO_3_ increases, the color of the CD solution goes from pale-yellow to brown, and the production yield of CDs increases. The DCE has two important functions in the reaction: to dissolve the WEPS, and to form a network of three-dimensional crosslinks by linking the benzene rings of polystyrene in the presence of HNO_3_. The central emission wavelength changes from 470 to 630 nm, with 5 to 30 μL HNO_3_ variation, and the FWHM almost covers the entire visible region [79]. In this case, the red shift is due to the increased degree of oxidation, which may reduce the bandgap of CDs [80]. The CDs also show an excitation-dependent photoluminescence, and the reported QY is about 5%.

Another mechanism that makes tunable fluorescence possible is owed to the coexistence of multiple emissions in the same material, as illustrated in Figure 6. Zhang et al. [47] demonstrated that tunable photoluminescence is ascribed to the ratio between the emissions attributed to the core states and those due to the surface states. They synthesized SSF N-doped CDs through microwave treatment of a solution of trisodium citrate dihydrate and a variable amount of urea. In addition to the emission at 440 nm, attributed to C=C bonds (core states), there is another contribution due to C=O and C=N bonds (surface states) that shifts the emission peak to longer wavelengths (500 nm) with increasing nitrogen concentration.

In this case, the SSF is owed to the Na ions, which attach to the surface of CDs, thus preventing aggregation of the cores. As a consequence, by introducing Na ions, and regulating the nitrogen and sodium at a suitable ratio, leads to high photoluminescence QY of over 80% [47].

A very similar result was obtained by Wang et al. [81], who achieved similar optical characteristics by synthesizing CDs from cyclen (nitrogen source) and citric acid in water solution by microwave irradiation. In this case, these authors performed a dual-step treatment, adding water after the first irradiation, and then irradiating again until solid-state products remain at the bottom of the beaker. Two emission peaks at 422 and 544 nm confirm the above theory that two emission mechanisms coexist, and are related to core and surface states, respectively. A highly unexpected result came from QY comparison between solid-state and aqueous solution of the same CDs. A QY of 48% is reported from SSF, against 20.7% from the liquid solution [81]; therefore, SSF CDs defeat self-quenching, showing also higher emission efficiency.

Zhou et al. [82] found a fabrication method focused on the realization of low characteristic luminescence lifetime solid-state CDs for communication applications. They performed a microwave treatment of citric acid and ammonia water. The product was dissolved in water and centrifuged to remove large-sized nanoparticles, then freeze-dried, dissolved in ethanol, and centrifuged again. After that, an oxygenation process with H_2_O_2_ was carried out to obtain the so-called “ox-CDs”. CDs present blue emission in water solution, but in solid state only ox-CDs emit a 525 nm green light under UV excitation, with 25% QY for 450 nm excitation wavelength. This oxygenation not only makes SSF possible by passivating surface states, but also reduces photoluminescence lifetime [82].

All the above-reported works share a common theme: CDs in liquid solution emit at shorter wavelength compared to their solid-state forms, and a red shift always occurs during this transition. Typically, CD cores have a bluish emission spectrum, so the functionalization and passivation of surface states, on one hand, helps to separate these cores from each other, but on the other hand, it introduces other lower energy states responsible for the SSF red shift. What can be seen as a limit to the development of SSF CD phosphors is instead an advantage, because yellow phosphors with a blue absorption spectrum are highly requested, especially in the field of color conversion WLED.

## 6. Color Conversion CD-Based LEDs

The first attempt to fabricate an LED based on color conversion of CD dates back to 2013. At the time, scientists considered that CDs could be mixed with other well-trialed color converters (such as QDs) in order to improve the optical characteristics of the resulting LED [83,84,85]. Typically, CDs emitted in the blue region, so the result of mixing them with other color converters produced a white light.

### 6.1. AIQ and Possible Solutions

One of the first problem that scientists faced when trying to use CDs for LED fabrication was the degradation of CDs that, like any other organic converter, limited the potential of CDs in emitting devices. Better results were obtained by embedding the CDs in ionic salt, such as KBr and NaCl. Other embedding materials provided lower performances [86]. Many other materials were used as a matrix to embed the CDs. Among them were PVA [87], PMMA [85], epoxy resin [88], silicone [89], and many others. An alternative matrix to embed CDs is the Metal–Organic Framework (MOF); MOFs are known as networks with organic ligands containing potential voids, which can be fitted to the extremely small size of CDs, reducing their self-quenching-related problems in composite materials. Lanthanoid MOFs (Ln-MOFs) containing blue-emitting CDs (CDs@Ln-MOF) have been used for WLED fabrication, and enrich a new approach for multicolor lighting application (e.g., anticounterfeiting inks) [90]. However, by improving the synthesis techniques and the choice of the precursors, CDs have turned out to be more stable than other organic materials and dyes; moreover, they suffer from the AIQ effect. This detrimental effect can be reduced, for instance, using intercrossed carbon nanorings (IC-CNRs) with very pure hydroxyl surface states. This structure enables them to overcome AIQ effects [91]. In other cases, the photoluminescence self-quenching can be efficiently suppressed if CDs are embedded within a polyvinyl pyrrolidone (PVP) matrix. Furthermore, a flexible solid-state material, such as organosilane (OSi), used to incorporate red-emitting CDs, has been proven to prevent the AIQ effect in solid-state CDs [92]. Additionally, PVA [93] and silica [94] have been proven to prevent AIQ. In particular, in PVA, the photoluminescence quenching is avoided due to the hydrogen bonds between the CDs and PVA molecules [95].

The nitrogen source used as a precursor can be carefully chosen in order to also serve as a surface passivation agent; this reduces or avoids the AIQ effect in the CD preparation. For example, Liu et al. used citric acid as a carbon source and branched poly(ethylenimine) (b-PEI) as a nitrogen source. The b-PEI also acts as a surface passivation agent, and avoids the luminescence quenching. Different reactions take place between b-PEI and other carbon sources (instead of the citric acid precursor), and this could generate new multicolor-emitting CDs [96].

AIQ can be prevented or reduced in LEDs using an appropriate matrix as an encapsulant. For instance, the introduction of the silane coupling agent (KH-792) has been proven to suppress fluorescence quenching [97]. Positive results were also obtained using MOF, as reported earlier [90]. Other scientists exploited the properties of some copolymer precursors, such as poly(methyl methacrylate-co-dimethyl diallyl ammonium chloride) (PMMAco-DMDAAC), which can easily become honeycomb-patterned in a chloroform solution. Depending on the molar ratios of DMDACC to MMA (1:5, 1:10, 1:15, and 1:20), the resulting copolymers are indicated as PMD-5, PMD-10, PMD-15, and PMD-20, respectively. Using these matrices to embed polymer CDs for LED fabrication, these authors obtained an increase in LED luminous efficiency up to 41.9%, compared to that of the non-patterned LED. In general, the smaller the molar ratios of DMDACC to MMA, the smaller the honeycomb size; therefore, the highest luminous efficiency values are measured for the smallest molar ratios of DMDACC to MMA (1:5), since the patterned matrix efficiently tackles the AIQ effect [98].

Another material used as a matrix to prevent the detrimental AIQ effect is the zinc–borate matrix [99].

In a certain type of CDs, AIQ can be transformed in an enhancement of fluorescence emission by the opposite mechanism, called “aggregation-induced emission” (AIE). The presence of KCl, together with citric acid and L-cysteine as precursors, causes the CD’s blue fluorescence to be suppressed by AIQ, while longer wavelengths result in the enhancement of yellow emission. This phenomenon, which is well investigated for other dyes and organic fluorescent materials (see for instance, the exhaustive chapter reported in [100]), can be explained with reference to Figure 7. When the CD concentration increases, the CDs assemble into aggregates. When KCl is one of the precursors, together with citric acid and L-cysteine (CD1), the residual blue-emitting CDs (at 450 nm) can transmit their energy to the large aggregates of CD1, which are yellow-emitting (right side of Figure 7a). In the meantime, the reabsorption of the blue light by the agglomerated CD1 (at 470 nm) generates an enhancement in yellow emission with increasing concentration (Figure 7a). Without using KCl as a precursor (CD2), the aggregated CDs do not emit any fluorescence (right side of Figure 7b). Instead, since the absorption band of the aggregates is broad, the energy of the residual CD2 transitions is non-radiative (Figure 7b). The increase in concentration leads to the luminescence quenching [99].

### 6.2. Chromaticity and Luminescence Properties

During the years 2005–2015, the CDs fluorescence reported in all the published works always fell within the blue–cyan region, unless CDs were mixed with other phosphors or QDs. Sun et al. [89] succeeded in obtaining a high CRI WLED using only organic QDs; however, they used CDs for blue emission and polymer dots for green and red colors. Afterwards, some groups discovered that it was possible to tailor the CD color emission (from blue to red) simply by varying the CD concentration in the host matrix, and thethickness of the conversion layer made with this CD-based matrix [102,103]. In particular, the same effect is obtained by varying the CD concentration in the matrix while maintaining the same film thickness.

Until 2017, most of CDs produced for optoelectronic applications emitted in the blue region. WLEDs based on CDs as color converters were demonstrated either by exploiting the broad emission band of CDs, generally centered at 450–460 nm (high correlated color temperature (CCT)), or by mixing the blue-emitting CDs with other types of green/red-emitting phosphors, such as QDs and Ce:YAG (low CCT). In 2017, Chen et al. [104] used orange-emitting CDs to fabricate WLEDs. The precursors used were *p*-phenylenediamine (p-PD) as the carbon source and formamide (FA) as the nitrogen source. Different emission peaks could be obtained by controlling the surface states during the solvothermal fabrication process. Controlling was carried out by adjusting the temperature and reaction time, resulting in two color emissions: blue and orange. In order to increase the emission wavelength of CDs, Lin et al. [92] observed that at high reaction temperatures (100–200 °C for 6 h), the surface oxidation becomes more severe (using citric acid as the carbon source and urea as the nitrogen source); this results in higher oxygen and nitrogen content, which is responsible for longer wavelength emission.

Another way to obtain long-wavelength-emitting CDs is to exploit the “solvatochromism” in CD solutions by simply varying the solvent polarity [105]. However, solvatochromism has resulted in the fabrication of a wide color palette of emissive CDs. Two common precursors, such as citric acid and urea, allow the attainment of a large color gamut simply using three different solvents (water, glycerol, and DMF) [106]. The solvents control the dehydration and carbonization processes of precursors, resulting in the creation of CD sp^2^-conjugated domains of different sizes. This mechanism causes the CDs to emit at different wavelengths, ranging from blue to red.

For instance, solvothermal methods are also used to fabricate CD-based green LEDs, pumped by blue InGaN LEDs (450 nm) [107]. Additionally, Yuan et al. [108] observed multicolor luminescence from CDs prepared with the same precursors and dissolved in different solvents (ethyl acetate, ethane-diamine, oleylamine, dimethyl sulfoxide (DMSO)); in this case, the emission wavelength can be tuned from 475 to 624 nm, i.e., from blue to red.

Nowadays, emission of a wide palette of full-color CDs is made possible by changes in surface functional groups, such as C=O and C=N. Different surface groups can be prepared by varying the ratio of the precursors; various surface states correspond to a relatively wide distribution of different energy levels, and consequently to adjustable full-color emissions from CDs [109].

Generally, for solvothermal processes, blue-, green-, and red-emitting CDs are based on different absorption mechanisms of light pump: the absorption peaks are attributed to π→π* and n→π* transitions of carbon cores for blue-emitting CDs, to C=O groups on the surface state for green-emitting CDs, and to C=O and C=N groups on the surface state for red-emitting CDs [110].

Multicolor emission can be achieved (and successfully exploited for WLED fabrication) by the triple emission of CDs with a multiple core@shell structure. Different emissions should correspond to different energy levels from the core band, edge band, and surface band [111]. A model that can potentially explain the three emissions and their origin is sketched in Figure 8.

Due to the large gamut of CD emissions, dichromatic and trichromatic WLED can now be realized [112]. These WLEDs are fabricated by simply controlling the solvent polarity, which affects the CD’s fluorescence wavelength. The phenomenon is correlated to the way the surface states of the CD are linked with solvent molecules (Figure 9a) that are responsible of different emission wavelengths, and explains the bathochromic shift in the emission (i.e., the change of the spectral band position). An analysis of HOMO–LUMO energy levels demonstrates that the bandgap of CDs decreases with the increase in solvent polarity (Figure 9b).

Until few years ago, red-emitting CD synthesis was extremely difficult; there has since been much progression, and in 2018, enhanced red emissive CDs, with QYs as high as 25%, were common to prepare. In this case, the red emissive CDs were embedded in a matrix of PVP. The protection of PVP allows CDs to preserve a strong luminescence under UV excitation, and for a long time, providing a high device stability [113].

Chromaticity variation in CDs can also be obtained by preparing the dots at different precursor mass ratios. In fact, by varying this ratio, CDs exhibit different element contents and functional group contents. Particle size depends on the mass ratio, as well as the emissive properties. In particular, it has been shown that a variation in the mass ratio, which induces a decrease in CD size, also induces a red shift in the emission wavelength. This can be helpful for fabricating CD-based WLEDs with tunable light-emitting colors [114].

Another way to prepare multicolor emissive CDs is to exploit the isomers of a precursor, such as the phenylenediamine, i.e., o-phenylenediamine (o-PD), m-phenylenediamine (m-PD), and p-phenylenediamine (p-PD). These isomers have similar chemical structures, but the phenyl groups are differently located. This property makes them ideal precursors for producing multicolor emissive CDs. In fact, phenylenediamine isomer-based CDs, simply prepared by microwave heating, possess a plethora of surface states, such as amine, aliphatic C−H, amide carbonyl, C=O/C=N, C=C, C−N=, C−N, and C−O. When the content of C=O/-CONH- and the ratio of C=O/C−O both increase, the luminescence gradually moves towards the red spectral region, from m-CDs, o-CDs to p-CDs. Similarly, by increasing the content of N element and non-amino N, the QY increases gradually from p-CDs, m-CDs, to o-CDs. Eventually, the increase in C=O/-CONH- initiates the red shift in the emission, while the increase in non-amino N enhances the fluorescence intensity and the QY. This is explained by the sketch in Figure 10, where CDs are embedded in starch [115].

In N-doped CDs, the presence of C=S bonds, together with C=N bonds, are responsible for a red shift in the emission, as illustrated in Figure 11.

The introduction of the N- and S-doped states, starting from the appropriate precursors, allowed Kumari et al. to prepare orange emissive CDs, and, more generally, multicolor emissive CDs, which were efficiently used for the fabrication of orange and white LEDs [116].

A more complete study shows the influence of temperature in solvothermal approaches on the luminescent properties, notably, the wavelength emission and the QY [117]. These mechanisms are summarized in Figure 12.

An increase in temperature promotes the escape of N and O elements from the carbon nucleus, resulting in improved surface functionalization. This mechanism also explains why the QY increases with temperature. An additional effect is the higher degree of oxidation (and passivation) with temperature. The enhanced surface passivation leads to the emission wavelength shift from blue to red. Blue CDs are rich in O–H surface groups and an increase in temperature promotes the formation of surface carboxyl carbons (COOH) groups, moving the emission spectrum towards the green region. Further increases in temperature cause the emission wavelength to shift towards the red region of the spectrum; the surface states are now rich in amino groups NH_2_. It is worth pointing out that nitrogen doping of CDs corresponds to CDs having a new energy level (N-state). The presence of this new level reduces the energy bandgap, as shown in Figure 12. One can see that the single-energy N-state splits into more energy levels with the beginning of nitrogen doping; this further reduces the energy bandgap ΔE_v_.

Nevertheless, the change in CD color emission seems to be correlated with the sizes of CDs and, consequently, with the pH of solutions. Measurements carried out by transmission electron microscopy (TEM) on three kinds of CDs prepared by solvothermal methods confirmed this correlation (Figure 13a–c). The size modifications occur after tuning the pH of the CDs solution. The size distribution histogram (Figure 13d–f) related to blue CDs (B-CDs), green CDs (G-CDs), and orange CDs (O-CDs), displays average CD particle sizes of about 0.9, 1.7, and 3.6 nm, respectively [118].

A larger CD size corresponds to a lower bandgap, so the difficulty to synthesize red- emitting CDs lies in enlarging of the dot sizes. The challenge is that CDs are less reactive with size, making large-sized CDs difficult to produce. Thus, it is necessary to find a precursor sufficiently reactive to produce high color purity long-wavelength emission. For instance, resorcinol is a suitable precursor to achieve pure red CDs, and it is also able to obtain pure green CDs, simply by shortening the reaction time [12].

Note that there are other surface functionalizing groups that can change the color of the emission and the QY of CDs. This is very often related to the nature of the solvent. For instance, organosilane-functionalized carbon dots (Si-CDs) have been produced. The surface of CDs contains N-groups including C=N and C–N, and Si-groups including Si–O–C and Si–O–Si. If dispersed in acetone, Si-CDs are blue emissive, while in solid state they show yellow fluorescence. An interesting property of Si-CDs is that they behave as selective fluorescent “ON–OFF–ON” nanoprobes for chromate (Cr(VI)) and ascorbic acid (AA). Whereas they are quenched by the presence of Cr(VI), the addition of AA allows fluorescence to be restored (because of the reduction of Cr(VI) by AA) [119].

CD color emission depends not only on the kind of precursors, but also on the molar ratio between the different carbon, nitrogen, and other sources. Su et al. reported multicolor zinc-atom-doped CDs (based on p-PD and ZnCl_2_) fabricated by varying the molar ratio of the precursors. They achieved red emissive CDs at a 1:0 molar ratio, purplish-red emissive CDs (1:0.1), purplish-blue emissive CDs (1:0.5), and blue emissive CDs (1:1) [120]. Varying the proportion of the precursors, and also the types of solvents, the gamut of the achievable colors widens more and more [121].

In a solvent-responsive process, multicolor CDs are obtained by dispersing the CDs in immiscible solvents, providing a full range of visible colors. The fluorescence emission in immiscible solvents is red-shifted by enhancing the polarity, as reported in Figure 14.

Nowadays, much progress has been made in the preparation of multicolor CDs. Nevertheless, the aqueous solution employed in hydrothermal methods hinders the production of both red- and near-IR (NIR)-emitting CDs. Thus, solvothermal methods should be used to fabricate NIR-emitting CDs, where aromatic ring structures are often present, causing these CDs to be toxic. In order to prepare non-hazardous NIR emissive CDs, citric acid and thiourea have been used as carbon sources, while ammonium fluoride was used as the dopant [123]. NIR emissive CDs are formed due to the rather highly negatively charged, electron-withdrawing group (F^−^), which lowers the bandgap. In addition to the F^−^ group, NIR-emitting CDs have many other surface functional groups, and a large conjugated sp^2^ core structure.

### 6.3. QY Improvement

In order to increase the photoluminescence QY of CDs, Li et al. [124] exploited the phosphorescence of N-doped CDs. In particular, C=N bonds on the surface of N-doped CDs promote the formation of triplet excitons through n−π* transition, which is considered as the origin of phosphorescence. A long phosphorescence lifetime of 1.06 s was achieved. Therefore, the whole QY is the sum of the fluorescence QY (29%) and the phosphorescence QY (7%). Furthermore, the matrix used to incorporate N-doped CDs (formed by melting recrystallized urea and biuret by heating urea) suppresses the vibrational dissipation of triplets due to the considerable rigidity of melting recrystallization urea combined with hydrogen bonding between biuret and NCDs. This principle is illustrated in Figure 15.

Fluorescence and phosphorescence occur together in carbonized polymer dots (CPDs). They are a new type of CD that link the benefits of both facets of composite-based CDs materials (excellent emissive properties) as well as the matrix effects typical of polymers (covalent bonds substituting for supramolecular interactions). Their preparation is based on a one-step process involving polymerization, deamination, and dehydration reactions of urea and phosphoric acid in an aqueous environment. These CPDs emit a dual-component white light involving two simultaneous processes of fluorescence (S_1_→S_0_) and phosphorescence (T_1_→S_0_). WLEDs based on CPDs show outstanding stability after continuously working for 144 h, and a luminous efficacy as high as 18.7 lm/W, under a 370 nm-wavelength excitation [125].

Another parameter that influences the QY and the performance of CD-based WLEDs is the pH of the solution, in the case of preparation by hydrothermal treatment. For instance, Lu et al. [126] proved that an acid solution produced CDs with a high QY. Moreover, by increasing the temperature of the process (from 200 °C to 300 °C) supersmall CDs (0.5 nm of diameter) were produced, having a strong white photoluminescence. WLEDs with CRI and CCT of 81 and 6786 K, respectively, were fabricated using the frequency down-conversion properties of these supersmall CDs. A schematic illustration of the different structures obtained by varying pH and temperature with this method is reported in Figure 16. It is worth pointing out that white luminescent supersmall CDs are produced starting from an aqueous solution of L-serine and L-tryptophan at 0.5 pH value.

Higher values of QY can also be obtained by appropriate tailoring of the CD surface-states. Additionally, the choice of solvent and reaction temperature influence the QY value. For instance, yellow CDs, with surfaces made of pure starch particles, have been proven to enhance the QY from 10.4% to 23.1% when changing the reaction solvent from ethanol to deionized water, and increasing reaction temperature from 180 °C to 200 °C [127].

Doping is another way to improve the QY of CDs, especially in the longer wavelength region. Many metal ions were studied as dopants in orange emissive CDs prepared by solvothermal methods. For Fe^2+^-, Co^2+^-, and Zn^2+^-doped CDs, the QY reaches values of 18.68%, 10.81%, and 20.65%, respectively. The best results were achieved with Mn-doped CDs, which exhibited a record QY value of 28.5 [118].

Surface state modifications can also improve the QY. Li et al. proved that polymer surface modification enhances the luminescent properties of CDs. Non-functionalized CDs obtained by one-step pyrolysis starting from citric acid, and tri(hydroxymethyl) amino methane hydrochloride (Tris-HMA) as precursors, exhibit a 12% QY, while the same CDs decorated by hydrophobic polystyrene (PS) through Schiff base condensation/Michael addition reaction (CD@PS) and catechol-terminated with hydrophilic poly(poly(ethylene glycol) methyl ether methacrylate (PPEGMA) (CD@PEGMA), shows a QY value of 15% for CD@PS and 23% for CD@PEGMA [128].

As usual, bare CDs show dangling bonds on the surface. Dangling bonds are a possible source of non-emissive trap states on the surface that, in turn, are responsible for a decrease both in QY and in fluorescence. As reported earlier, surface state modifications can provide the required conditions to occupy a dangling bond site. In [129], where CPDs are formed by carbonization of citric acid and octadecene, surface states are passivated by hexadecyl amine (HDA) as the surface functionalizing agent. In particular, HDA provides −NH2 groups to the CD surface, enhancing QY, color purity, and CRI.

### 6.4. Luminous Intensity and Luminous Efficiency

One weakness of CDs used as color converter in an LED is the fact that CDs are photoluminescent in liquid solution, especially in water, which is one of the most polar solvents. In a solid state, the AIQ effects have been posing a barrier to their widespread use in LED manufacturing. Unfortunately, most of matrices that have been proven to embed CDs without strong AIQ effect are not soluble in water. The incompatibility between aqueous solutions of CDs and silicone-like matrices can be resolved by adding ethanol (which dissolves silicone) into the aqueous mixture [130].

Another weakness of CD-based LEDs is the low values of luminous intensity and luminous efficiency. In inorganic phosphor-based WLEDs, a well-known technique to improve spatial uniformity and efficiency is to add mineral diffusers, such as TiO_2_, CaF_2_, and SiO_2_, to the encapsulant [131]. The difference between the refractive index of the diffusers and of the encapsulant causes scattering from the diffusers, which uniformizes the chromaticity distribution of the emissive radiation. This has also been confirmed by working with a remote phosphor WLED configuration: using TiO_2_ as the encapsulant, the luminous flux significantly improved by 8.65% [132]. The same effect has been tested with CDs and TiO_2_ nanoparticles dispersed in a silicone encapsulant; here, the luminous intensity is increased by 31% for a concentration of TiO_2_ nanoparticles of 0.05 wt%. Unfortunately, the luminous intensity strongly depends on the concentration of TiO_2_ nanoparticles and, for a relatively high concentration, the luminous intensity decreases to unacceptably low levels (less than 10 mcd for a concentration of TiO_2_ nanoparticles of 3.25 wt%) [133].

The luminous intensity of CD-based LEDs can also be enhanced by exploiting the surface plasmon resonance from Ag nanoparticles [134]. In fact, metal-enhanced fluorescence (MEF) has been shown to improve photoluminescence intensity [135,136,137]. MEF consists of a local electric field effect and radiative decay rate effect. The latter increases photoluminescence intensity and QY. This mechanism is explained by the Jablonski diagram reported in Figure 17.

As shown, in the presence of metal particles, there are three decay paths (only two in free-space conditions). One of these decay paths, indicated as Γ_m_, corresponds to radiative rate in the presence of metal, and it is summed to the standard radiative rate factor Γ. This corresponds to an improvement in the total radiative emission rate. MEF was also successfully exploited by mixing CDs with colloidal Au nanoparticles. It has been demonstrated that the luminous intensity of CDs can be increased by a factor of three [138]. Cui et al. proved a further improvement by a factor of five by coupling blue CDs (embedded in PVP) with an Ag film made of surface plasmonic resonance “islands” [139].

The luminous intensity and its spectral distribution depend on whether trapping states are passivated or not. Hence, one way to increase the luminous intensity is to passivate the CD surface states. Generally, as reported in the self-explanatory Figure 18, unpassivated CDs photoluminescence is dependent on the excitation wavelength, while the passivation of surface states allows not only excitation-independent behavior, but also a more uniform spectral distribution and a higher quantum yield [140].

Carbon dots–silica spheres (CS) composites show a fluorescence intensity 10 times higher than that of CDs alone. Using citric acid and EDA as precursors, the absolute fluorescence QY values of CDs and CS composites prepared by hydrothermal method are 4.57% and 40.1%, respectively. Therefore, there is a 10-fold enhancement both in luminescence intensity and in QY [141].

### 6.5. Stability

Although the above-mentioned struggles in using CDs for LED color conversion (more complex process for long-wavelength emissions, relatively low values of QY, poor luminous efficiency), compared to other organic converter or dyes, CDs seem to be highly promising as phosphors for LEDs due to their high photoluminescence stability. Generally, the Achille’s heel of dyes used for lighting purposes is their poor stability and photobleaching [142,143,144]. On the contrary, CDs have been proven to be highly stable over long periods of time, and also resistant to high currents (no changes in 72 h at 90 mA [145]). It was also found that, if dispersed in silicone rather than in a solution, the stability of WLED is considerably higher, since the excited singlet state is hard to annihilate in silicone. Therefore, the movement of CDs in the silicone matrix is prevented (while this does not happen in solution), decreasing the probability of collisions between excitons [140].

PVA as encapsulant was also found to provide high UV stability: minimal attenuation of photoluminescence intensity was recorded during 8 h exposure to 370-nm wavelength light (less than 10% for blue, green, and orange CD-based LEDs) [146].

Orange-emitting CDs, obtained by solvothermal reaction of 2, 7-dihydroxynaphthalene (C_10_H_6_(OH)_2_) and N, *N*-dimethylformamide (DMF, C_3_H_7_NO), embedded in KH-792, have a strong UV stability against photobleaching (after 60 min of 365-nm wavelength irradiation, the emissive intensity is only reduced of 4.7%). Their thermal stability is good as well (no weight loss up to 200 °C) [97]. Better results are achieved by exploiting the surface plasmonic resonance of an “islands” Ag film coupled with a blue CDs/PVP solid film. Here, the stability increases by a factor of 10, with reference to the previous orange-emitting CD case: the emission intensity decreased by about 5% after 10 h under continuous operation [139].

### 6.6. Biocompatibility

CDs are biocompatible ecofriendly materials: this is one of the reasons for their success in optoelectronics. Going forward, some research groups used biomaterials (also edible materials) as carbon sources to embed the fabricated CDs. For instance, potato starch was used both as a precursor to fabricate blue-emitting CDs and to encapsulate the CDs over UV LEDs used as a pump source. QY was rather low (about 2.5%), but it doubled when the CDs were N-doped using a nitrogen source such as EDA. Furthermore, if N-doped CDs showed another fluorescence peak with longer wavelengths, then a WLED could simply be fabricated by embedding the N-doped CDs into potato starch [147]. Better results were obtained by Liu et al. [148], who used starch as a carbon source for CDs, and achieved a maximum QY of 9.75% by optimizing the mass ratios of precursors.

Materials prepared from natural biomass can instead be used as encapsulants. For instance, transparent wood films embedding multicolor CDs have been fabricated by removing lignin from wood using deep eutectic solvent [149].

Waste expanded polystyrene (WEPS) can also be converted into solid-state CDs with good performance. In the presence of HNO_3_, solid-state CDs are produced. Their luminescence can be tuned from white to orange. FWHM of the emission spectrum ranges from 150 to 200 nm [79].

The sucrose contained in caramelized sugar can be used both for CD preparation and for their encapsulation. It was shown [150] that caramelized sugar-based CDs convert blue and green LEDs into white and yellow emission, respectively. It has been proven that the origin of radiative emission in caramelized sugar-based CDs is due to their surface states.

Steric hindrance, arising from the spatial arrangement of atoms, can be exploited to prepare biomass-based fluorescent CDs in solid-state form. The CD structure, derived from the use of dehydroabietic acid and ethanolamine as precursors, contains a graphitic carbon core, a sterically hindered non-planar tricyclic phenanthrene skeleton, and different functional groups, at the same time. The sterically hindered skeleton blocks π–π interactions between carbon cores, thus enabled emission in solid state [151].

Avocado peel and sarcocarp are also used to prepare blue emissive CPDs-P (peel) and blue–green emissive CPDs-S (sarcocarp) under an excitation wavelength of 365 nm. This is an great example of the use of biomass to fabricate CPD-based LED. Extremely stable warm and cool white LEDs are fabricated, as well as blue and blue–green LEDs [152].

From the point of view of circular bioeconomy, an interesting pathway to sustainability is traced by the paper of Xie et al. [153]. Herein, the authors extract the precursors for red CDs from spinach after soaking in ethanol/water (4:1 volume ratio) and further freeze-drying. Finally, the prepared CD-based LEDs are used for horticultural applications.

Eventually, with respect to inorganic QDs or rare-earth-based phosphors, CDs derived from biomass combines the advantage of low-price materials with the ease to achieve good optical properties. The production process of biomass-based CD converters is simplified, showing great potential in WLED applications.

### 6.7. Applications of CD-Based LED in Telecommunications

WLEDs based on CDs benefit of the short luminescence lifetime, which makes them suitable devices for high-speed visible light communication (VLC) systems. VLC based on WLEDs has attracted a huge interest in last years. Compared to traditional radio frequency communication, such as Wi-Fi and Bluetooth, VLC shows many advantages: unregulated communication spectrum, no RF interference, higher security, and environmental friendliness [154,155]. CDs, as color converters, exhibit shorter luminescence lifetimes than those of rare-earth-based phosphors, such as Ce:YAG. Liu et al. fabricated a high-speed VLC system with modulation bandwidth of 55 MHz and data transmission rate of 181 Mbps based on WLEDs, which exploit CDs as wavelength converter material. This system shows higher performance than those based on standard phosphors WLEDs. In fact, using citric acid and b-PEI as precursors for the CD preparation, a luminescence lifetime as short as 4 ns was measured [96].

### 6.8. Synoptic and Chronological Framework of Color Conversion CD-Based LEDs

In this paragraph a complete and exhaustive collection of all the published papers regarding CD-based LEDs is presented. Form early attempts in 2013 to the most recent results obtained today, Table 1 contains all methods and results reported in literature until now, with useful information like excitation and emission wavelengths, CIE, CRI and so on.

## 7. Conclusions

In this work, materials, technologies, characterizations, issues, and challenges concerning CDs-based LEDs, where CDs are used for color conversion, have been reviewed. All the information and details reported in this paper have been intended to be as exhaustive as possible, despite the vastness of the subject. After an introduction, aimed at defining the topics and contextualizing them in the current research scene (*Introduction*), the paper focuses on the materials used in LEDs for color conversion; in particular, phosphors, QDs, and CDs (*Materials for frequency down-conversion LEDs*). A troublesome issue such as AIQ is covered in a separate section (*CD photoluminescence quenching*), where the origins of quenching are explained, and some solutions to avoid it are reported. After a section dedicated to the different methods (top-down and bottom-up) of preparation of CDs (*CD preparation*), the paper deals with different ways to synthesize SSF CDs for LED applications (*CD fluorescence in solid-state form*). SSF CDs can be obtained in matrices and as self-quenching-resistant CDs. Afterwards, one reaches the core of the paper, that is a complete review of the results concerning multicolor LEDs and WLEDs based on CDs as color converters (*Color conversion CD-based LEDs*). Several issues concerning materials and techniques, drawbacks and benefits, and concrete results are reported in this section (AIQ, chromaticity and luminescence properties, QY improvement, luminous intensity and luminous efficiency, stability, biocompatibility, and telecommunications applications), completed at the end by a synoptical and chronological table of the results obtained in color conversion CD-based LEDs.

Finally, the results achieved by the global scientific community clearly show that much progress has been made since the first attempt at developing CD-based LEDs in 2013. Wider gamut and higher values of luminous efficiency have been attained over the past few years (although some results are debatable). The biocompatibility of most of the materials used makes CDs strong candidates to substitute the use of toxic or hazardous elements used in standard phosphors and inorganic QDs. However, it is desirable to increase the values of QYs, luminous intensity, and luminous efficiency in order to reach the full maturity of the devices, and hypothesize a drastic material change in the market of color conversion LEDs.

## Figures and Tables

**Figure 1 materials-15-05450-f001:**
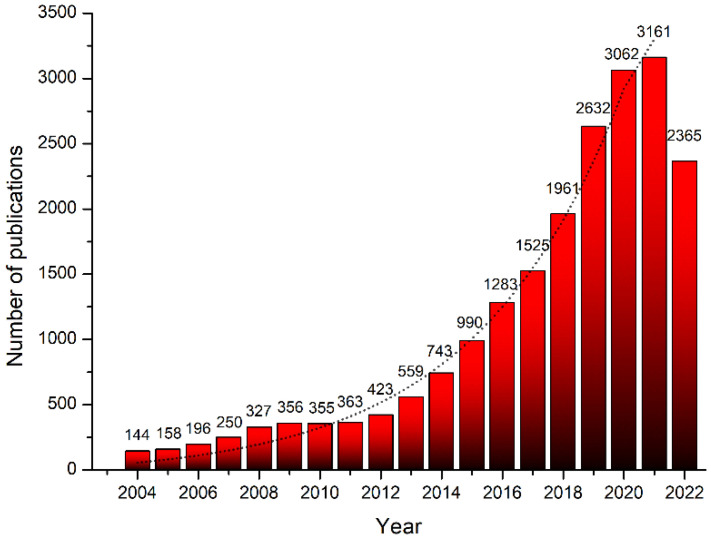
Trend of published papers related to CDs since their discovery. (Obtained using Scopus search results for “Carbon Dot”).

**Figure 2 materials-15-05450-f002:**
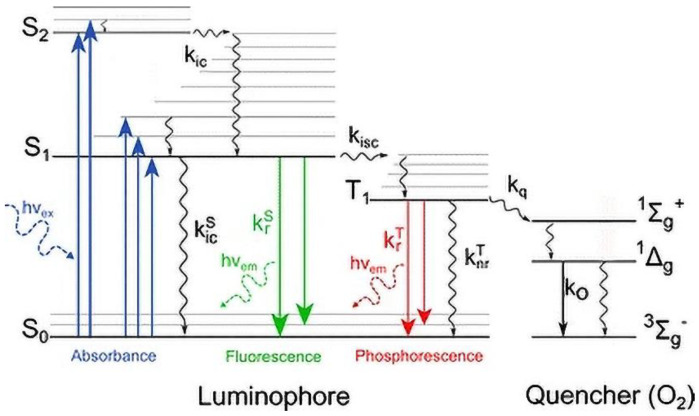
Semiconductor band diagram with charge transitions. (Reprinted from [23]; with the permission of the author).

**Figure 3 materials-15-05450-f003:**
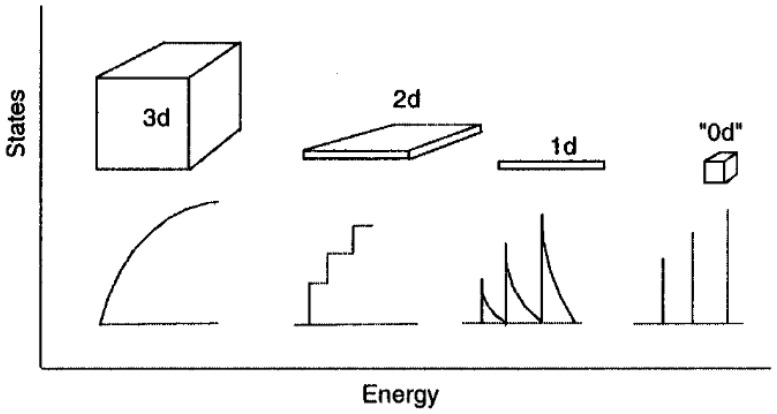
Idealized densities of state for one band semiconductor 3D, 2D, 1D, and 0D structures, indicated in the figure, respectively, as 3d, 2d, 1d, and 0d. In the 3d case (i.e., bulk material), the energy levels are continuous, while in the 0d or molecular limit, the levels are discrete. (Reprinted from [28]; with the permission of the American Chemical Society).

**Figure 4 materials-15-05450-f004:**
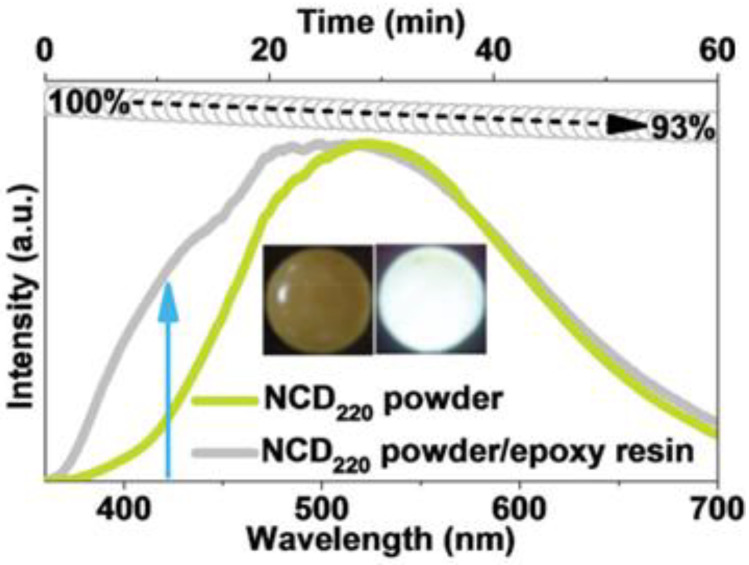
Fluorescence emission spectra (excited at 365 nm) of NCD 220 powder and NCD 220 /epoxy resin composite (solid lines), and time-dependent intensity (gray circles) of the composite continuously exposed to UV light (365 nm). Inset: the composite under daylight (left) and 365 nm of UV light (right). (Reprinted from [77]; with the permission of John Wiley and Sons).

**Figure 5 materials-15-05450-f005:**
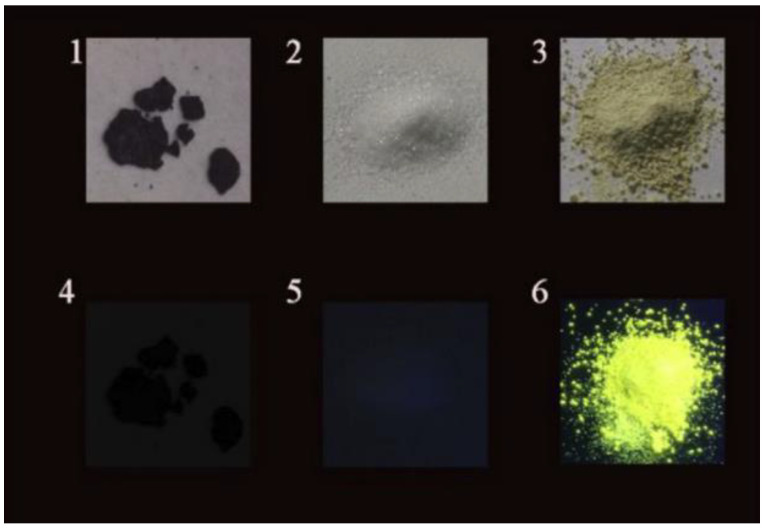
Photograph of the conventional CDs (**1**,**4**), trisodium citrate dihydrate (**2**,**5**), and NCD11 powders (**3**,**6**) under daylight (**upper row**) and 365 nm UV lamp (**bottom row**). (Reprinted from [78]; with the permission of Elsevier).

**Figure 6 materials-15-05450-f006:**
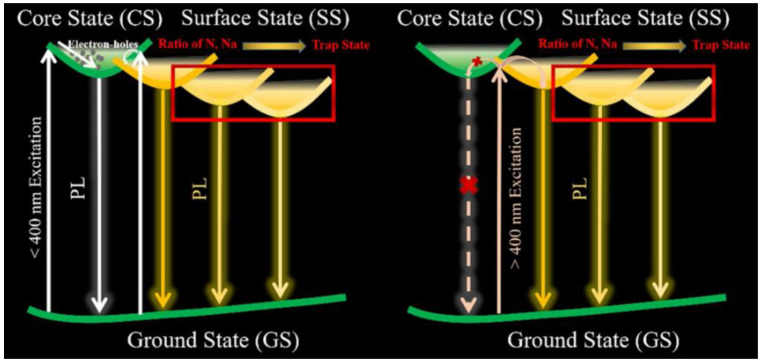
Mechanism representation of the dual fluorescence emission of solid-state N-doped carbon dots. (Reprinted from [47]; with the permission of Elsevier).

**Figure 7 materials-15-05450-f007:**
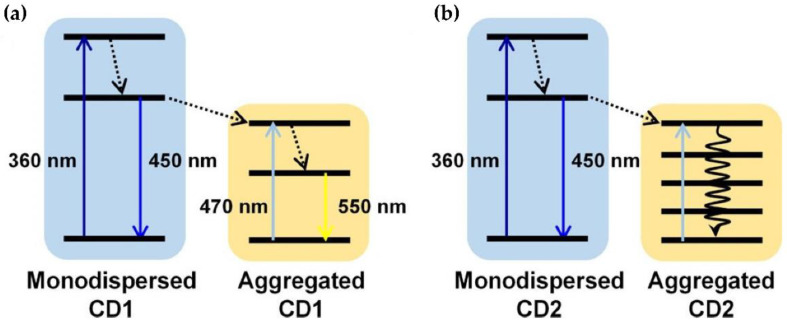
Energy band level alignments of monodispersed and aggregated CDs. Schematic of energy transfer that occurs in the aggregation process of (**a**) CD1 and (**b**) CD2. (Reprinted from [101]; with the permission of the American Chemical Society).

**Figure 8 materials-15-05450-f008:**
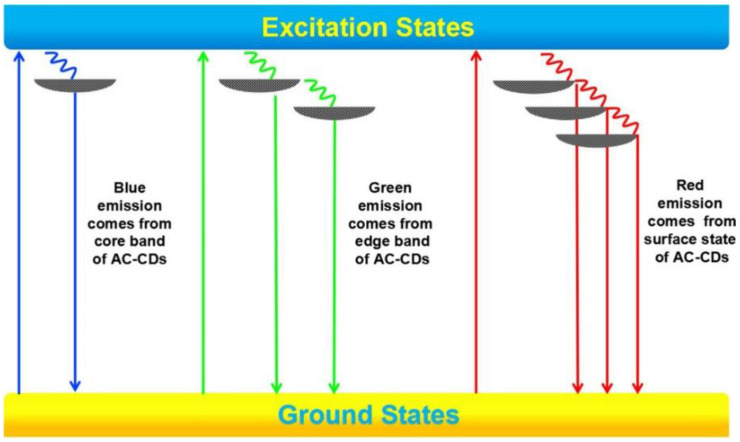
Energy level structures to explain the PL behaviors of the three different emissions from multicolor emissive CDs (indicated here as AC-CDs). (Reprinted from [111]; with the permission of the American Chemical Society).

**Figure 9 materials-15-05450-f009:**
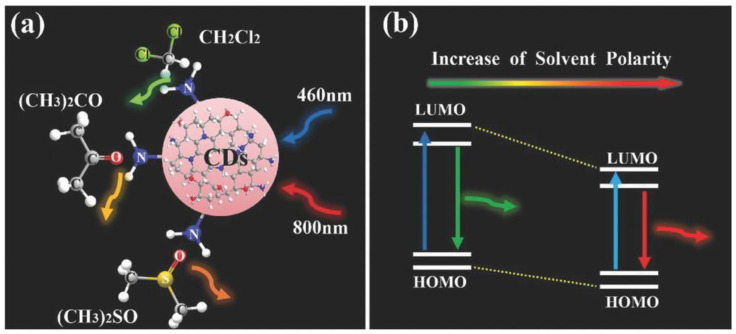
Schematic illustration showing the mechanism of multicolor emission from CDs. (**a**) The intermolecular interaction between solvent and CDs. (**b**) The change in energy levels of the CDs in different polarity solvents. (Reprinted from [112]; with the permission of John Wiley and Sons).

**Figure 10 materials-15-05450-f010:**
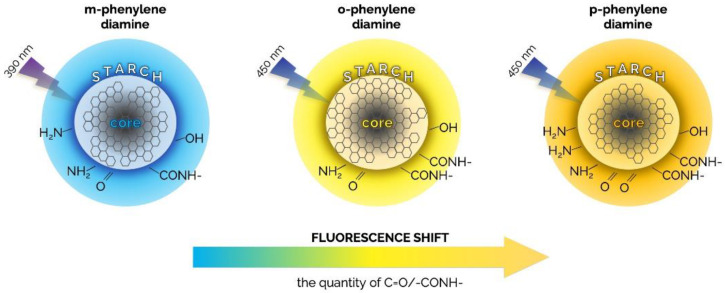
Sketch showing the fluorescence mechanisms of m-PD, o-PD, p-PD embedded in starch.

**Figure 11 materials-15-05450-f011:**
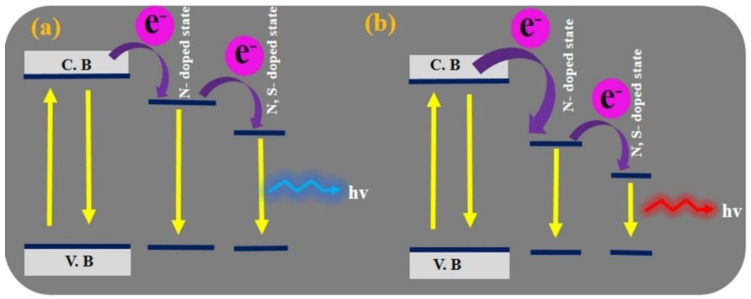
Illustration showing the energy states with different color emission by (**a**) m–CDs and (**b**) o-CDs. (Reprinted from [116]; with the permission of John Wiley and Sons).

**Figure 12 materials-15-05450-f012:**
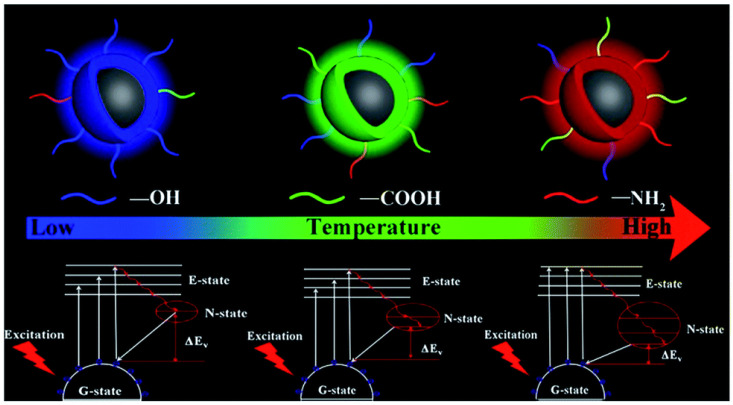
Schematic diagram of the luminescent and controllable synthesis mechanism. (Reproduction from [117]; with the permission of the Royal Society of Chemistry).

**Figure 13 materials-15-05450-f013:**
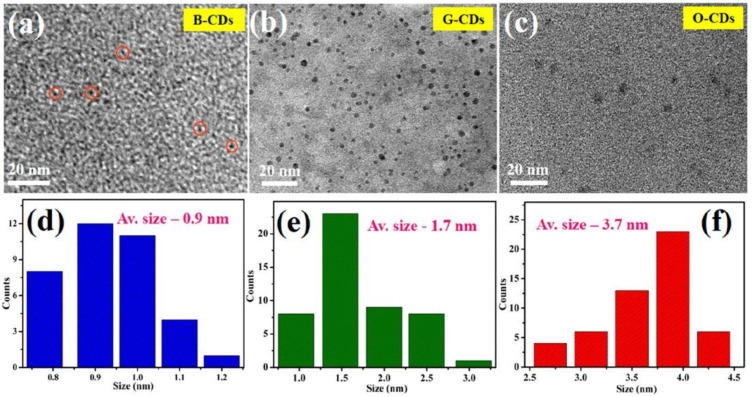
(**a**–**c**) HR-TEM images of blue, green, and orange color emissive CDs and (**d**–**f**) their size distributions histogram. (Reprinted from [118]; with the permission of the American Chemical Society).

**Figure 14 materials-15-05450-f014:**
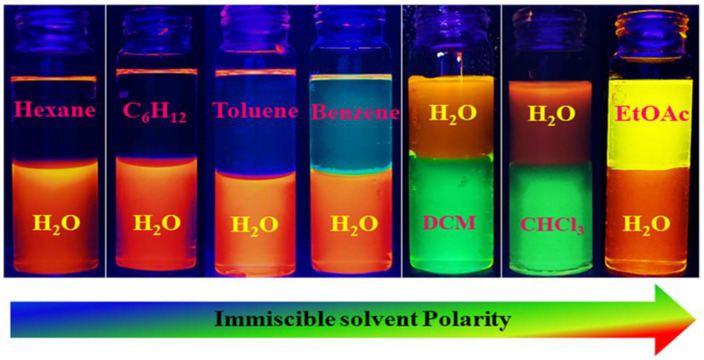
Two-color fluorescent layers of CDs in different immiscible solvents. (Reprinted from [122]; with the permission of the American Chemical Society).

**Figure 15 materials-15-05450-f015:**
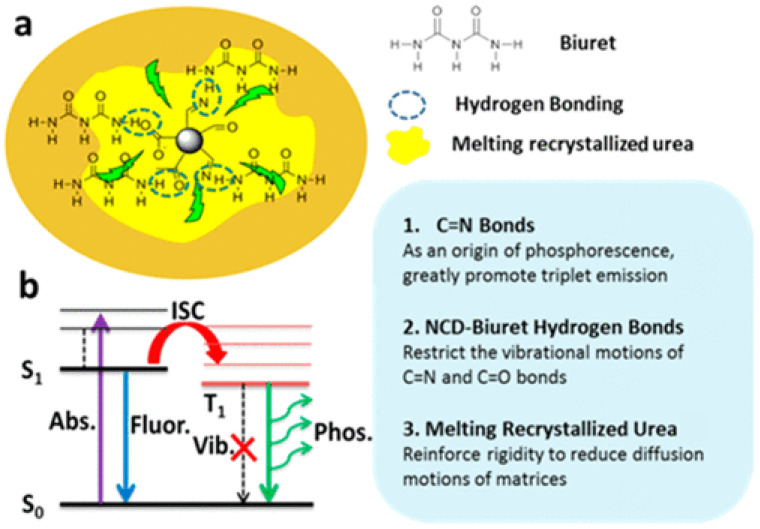
(**a**) N-doped CDs (NCDs in the figure) are embedded into melting recrystallized urea and biuret matrices. (**b**) Schematic illustration of possible energy structures of C=N bonds and phosphorescent emission processes. (Adapted from [124]; with the permission of the American Chemical Society).

**Figure 16 materials-15-05450-f016:**
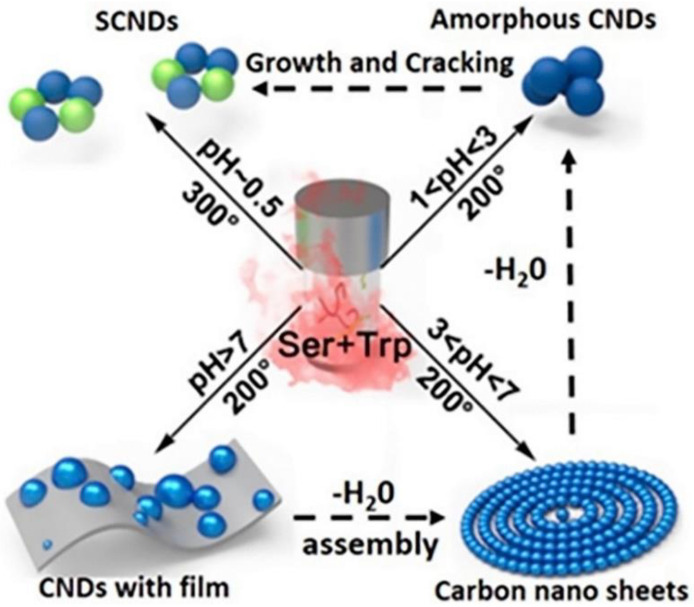
A schematic illustration of the preparation procedure of different structure carbon dots by hydrothermal carbonization of L-serine and L-tryptophan (Ser + Trp) at different pH values and temperatures. (Reprinted from [126]; with the permission of the American Chemical Society).

**Figure 17 materials-15-05450-f017:**
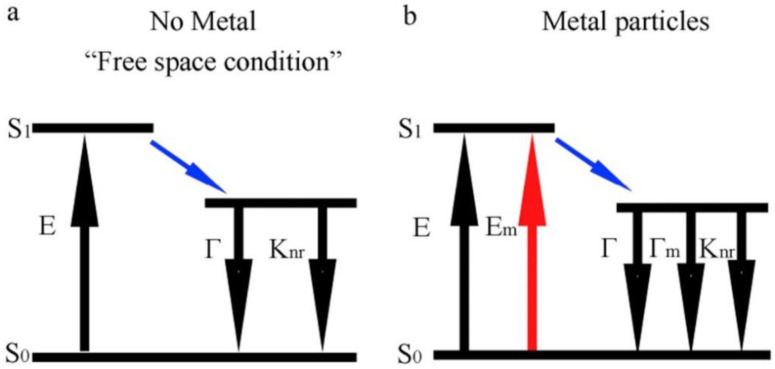
Classical Jablonski diagram for the free-space condition (**a**) and the modified form in the presence of metallic particles (**b**). E indicates excitation; E_m_ indicates metal-enhanced excitation rate; Γ indicates radiative rate; K_nr_ indicates non-radiative decay rates for excited state relaxation and Γ_m_ indicates radiative rate in the presence of metal. (Reprinted from [134]; with the permission of Elsevier).

**Figure 18 materials-15-05450-f018:**
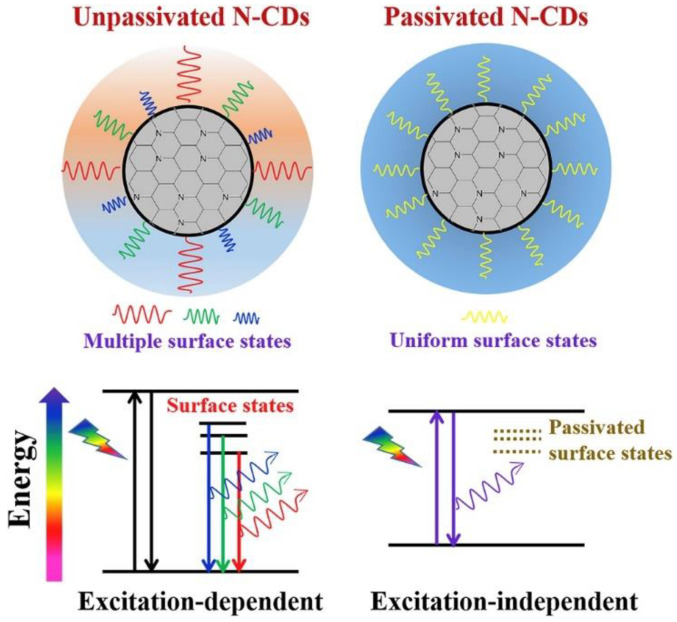
Visualized surface states of N-doped CDs with excitation-dependent and -independent behaviors. (Reprinted from [140]; with the permission of Elsevier).

**Table 1 materials-15-05450-t001:** Optical parameters, precursors, synthesis methods, and other parameters of CD-based color converter LEDs; (NR = not reported).

Year	Precursors	Preparation Method	QY (%)	LED λ_exc_ (nm)	LED λ_emis_ (nm)	CIE (*x*,*y*)	CCT (K)	CRI	Luminous Efficacy (lm/W)	Encapsulant	LED Emission Color	Notes	Ref.
2013	acrylamide, cadmium chloride, N-acetyl- L-cysteine (NAC), sodium borohydride, tellurium powder, sodium hydroxide	plasma-induced method (150 W)	~6%	380	490 (FWHM: 110 nm)	(0.20, 0.18) CDs (0.38, 0.36) mix	NR	87	30 (@350 mA)	silicone (OE-6550A: OE-6550B)	blue (CDs), white (mix)	CDs mixed with CdTe QDs	[83]
2014	citric acid, (3-aminopropyl) triethoxysilane (APTES) (1:1)	hydrothermal method (180 °C, 4 h)	20	380	broadband (two peaks at 470 and 550)	(0.27, 0.32) mix	9051	71	NR	silicone	white (mix)	CDs mixed with CDs nanocrystals	[84]
	citric acid, AAPMS	heating from a solution (240 °C, 1 min)	NR	400	490	(0.22–0.24, 0.38–0.43)	NR	NR	NR	silicone	bluish white	CDs embedded in silica, KCl, KBr, NaCl	[86]
	poly(acrylic acid) (PAA), glycerol	one-step pyrolysis	9	380	broadband (two peaks at 440 and 470)	(0.27, 0.32)	NR	NR	NR	silicone	white	CDs dispersed in a thermocurable resin (silicone)	[156]
	[3–(2-aminoethylamino)propyl]trimethoxysilane (AEATMS), citric acid	pyrolysis (240 °C, 2 min)	NR	385	450	(0.321, 0.312) at 10 mA (0.351, 0.322) at 30 mA	3825–6452	93	NR	PMMA	cold and warm white	CDs with zinc copper indium sulfide core–shell QDs	[85]
2015	EDA	exothermic reaction between P_2_O_5_ and H_2_O	28.5	360 (48 W)	broad band	(0.22, 0.33) at 360 nm of excitation	NR	NR	NR	PVA	bluish white	single precursor; stability after 16 h of irradiation	[87]
	PVA	one-step hydrothermal process	NR	460	550	(0.28, 0.27)	NR	NR	NR		bluish white	intercrossed carbon nanorings (with relatively pure hydroxy surface states); AIQ is reduced	[91]
	N-(β-aminoethyl)-γ-aminopropyl methyldimethoxy silane (AEAPMS), citric acid	pyrolysis (220 °C, 10 min)	78	460	570	(0.32, 0.33)	~6190	78.9	58.1	epoxy-resin	white	CD-doped sodium borosilicate gel (CD-NBS gel)	[88]
	citric acid, EDA	hydrothermal method	NR	385	~460	(0.326, 0.343) at 50 mA	2805–7786	85–96	4.9	silicone	high CRI WLEDs	phosphors based on the combination of CDs and polymer dots	[89]
	organic acid, silane	decomposing organic acid in silane coupling agent	NR	460 (10 W)	broad band (two main peaks at 550 and 600)	(0.376, 0.374) for thickness of 400 μm	2500–10,000	NR	NR	epoxy matrix deposited on polystyrene substrate	white	different colors depends on the thickness of the film where CDs are dispersed	[102]
	AEAPMS, citric acid	pyrolysis (240 °C, 1 min)	47	385	455, 550, 575, 585, 610	(0.22, 0.22) (0.34, 0.43) (0.42, 0.51) (0.46, 0.50) (0,52, 0.46)	NR	NR	1.21 cd/A	PMMA	blue, green, yellow, orange, red	five monochrome LEDs obtained by varying the CD concentration	[103]
2016	glucose, polyethylene glycol (PEG) 200	one-step hydrothermal method	3.5	365	broad band centered at 544	(0.32, 0.37)	5584	NR	NR	epoxy resin	cool white light	glucose is first used as a carbon source	[157]
	EDA, tetra-acetic acid (EDTA), ethylene glycol(EG)	microwave-assisted hydrothermal method	NR	365	broad band centered at 550	(0.34, 0.38)	5078	84	NR	PMMA	white	CDs with different functional groups	[158]
	folic acid, chloridric acid	hydrothermal treatment	36 (phosphorescence + fluorescence)	360	NR	(0.213, 0.204) (0.338, 0.363)	NR	NR	NR	melting recrystallization urea and biuret from the heating urea	blue white	phosphorescent N-doped CDs (NCDs)	[124]
	L-serine and L-tryptophan (molar ratio 3:1) at different pH values	one-pot aqueous synthesis controlled by pH	16.3 (pH ≈ 8.1) 26.99 (3 < pH < 7) 46.83 (1 < pH < 3)	365	broad band (410−700)	(0.29, 0.31)	6786	81	NR	DMMA	cool white	crosslinked polymer carbon film (pH > 7, 200 °C) polymer carbon nanosheets (3 < pH < 7, 200 °C), amorphous carbon structures (1 < pH < 3, 200 °C) supersmall CDs (pH = 0.5, 300 °C)	[126]
	citric acid, ammonium hydroxide	ammonium hydroxide modulated hydrothermal method	40	360 (1 W)	broad band centered at 450 (only CDs)	(0.18, 0.19) CDs’ LED (0.33, 0.37) RGB white	5447 (at 100 mA)	95.1 (at 100 mA)	NR	silicone, ethanol	white	mix of blue emission CDs, green emission SrSi_2_O_2_N_2_:Eu, red emission Sr_2_Si_5_N_8_:Eu	[130]
	aminopropyl methyl polysiloxane (AMS), citric acid	one-step solvothermal method	16	460	590	(0.33, 0.28)	NR	66.6	14	AMS-CDs crosslinked silicone rubbers (SRs)	white	AMS-CDs have a dual role of luminescence and encapsulation layer	[159]
2017	*p*-phenylenediamine (p-PD), formamide (FA)	solvothermal method (200 °C, 1 h)	10–15 (orange CDs) 20–30 (blue CDs)	465 (350 mA)	480−780	(0.283, 0.246) - (0.470, 0.358) only CDs (0.323, 0.326) - (0.419, 0.376) CDs + Ce:YAG	14,570 - 2158 only CDs 5977 - 3089 CDs + Ce:YAG	73.3 (only CDs) 85 (CDs + Ce:YAG)	60–80	polyvinylpyrrolidone (PVP)	white	mix of orange-emitting CDs and Ce:YAG	[104]
	NR (CDs produced by Nanjing JANUS)	NR	40 (reported by CD manufacturer)	UV (*λ* NR)	433–600	vary in a large gamut with TiO_2_ nanoparticles concentrations (at a fixed CD concentration of 10 wt%)	from 3000 to 19,000	from 40 to 85	~1.4 (increased by 31% with TiO_2_ nanoparticles)	silicone (mixed in a CDs-chloroform solution)	from cool to warm white	TiO_2_ nanoparticles used for enhancing the light scattering ability of the encapsulant	[133]
	citric acid, urea (dissolved in N,N-dimethylformamide -DMF)	hydrothermal method	48.5	460	broad band	(0.4595, 0.3925) (0.03 g CDs in 4 g OSi)	2561 (0.03 g CDs in 4 g OSi)	92.6 (0.03 g CDs in 4 g OSi)	30.6 (0.03 g CDs in 4 g Osi)	N-(b-aminoethyl)-c-amin-o-propyl methyl dimethoxy silane (Osi)	warm white	red emission CDs (casted on Ce3+: YAG phosphor in glass (Ce: PiG)	[92]
	citric acid, AEAPMS, nitrogen	one-pot hot injecting method	24.5	460	broad band 500–700	(0.33, 0.35) (0.6 mm-thick)	5435 (0.6 mm-thick)	74.6 (0.6 mm-thick)	41.26 (0.6 mm-thick)	Ag nanoparticle solution	from bluish to neutral white	surface plasmon resonance from Ag nanoparticles enhances CD fluorescence	[134]
	o-phenylenediamine, pPD, N,N-dimethyl formamide (DMF) (solvent)	hydrothermal method	52.4	460	broad band	(0.3943, 0.3869) (CDs + Ce:PiG)	3722 (CDs + Ce:PiG)	83 (CDs + Ce:PiG)	66.17 (CDs + Ce:PiG)	polyvinyl butyral (PVB)	warm white (CDs + Ce:PiG)	CDs shift gradually from 520 nm to 630 nm increasing their concentration in solution or changing the solvent polarity (solvatochromism)	[105]
	citric acid, urea (solvents: water, glycerol, DMF)	solvothermal synthesis	30–40	395, 440	448, 515, 540, 570, 603, 622, 638	(0.34, 0.31) (WLED)	5048 (WLED)	82.4 (WLED)	8.34 (WLED)	silica	blue, cyan, green, yellow, orange, red, white	realization of full-color emissive CDs by exploiting the solvatochromism of three different (water, glycerol, DMF)	[106]
	1,3-Dihydroxynaphthalene, KIO4 (dissolved in ethanol)	dehydrative condensation and dehydrogenative planarization (DCDP) method and solvothermally treatment	53	365	430, 510, 620	(0.3924, 0.3912)	3875	97	31.3 (at 20 mA)	PMMA	warm white	three layers of blue-, green-, and red-emitting CDs	[145]
	citric acid, Silane Coupler KH-602, nitrogen	one-pot method using CTAB as the cationic surfactant, NaSal as the structure-directing agent, TEA as a catalyst, and BTEE and TEOS as the silica source	29.8	362	470, 612	(0.294, 0.280) (0.356, 0.343)	NR	85–86	NR	silica	white	dual-emitting CDs/CaAlSiN3:Eu2+-silica powder; applicability as a lighting source for plant growth	[160]
	citric acid, N-(2-aminoethyl)-3- aminopropyltrimethoxysilane (KH792)	one-step solvothermal method		450	530				60,96	polymer	green	CDs converting blue light to green light	[107]
	citric acid, EDA	one-step microwave- assisted hydrothermal method	75.96	365	weak peak at 442, strong peak at 572	(0.42, 0.40)	3416	NR	NR	silicone	warm white	N-passivated CDs show record QY values	[140]
	oil-soluble 1,3,6-trinitropyrene (TNP) (as C and N sources), various solvents: toluene, acetone, EA, DMF)	solvothermal synthesis	65.93	460	broad band between 540 and 610	(0.32, 0.31)	6300	NR	NR	PMMA	white	mix of long-wavelength emitting CDs and green phosphors; effect of the solvents on the CD emission	[161]
	citric acid, organosilane	one-pot pyrolysis method	25–55	~450	~550	(0.24, 0.28)–(0.31, 0.43)	5030 (for x = 0.33 and y = 0.21)	74 (for x = 0.33 and y = 0.21)	79.4 (at 350 mA)	polymerized silane prefunctionalized carbon dots (SiCDs)	white	SiCDs individually polymerized one-component system, drip-coated and bulk polymerized on GaN LEDs	[162]
	citric acid, AEAPMS, nitrogen	pyrolysis	39	460	broad band centered at 560	(0.33, 0.36)	5653	78.2	43.75	colloidal Au nanoparticles	white	gold-carbon dots (GCDs) with high luminescence	[138]
	ammonium citrate, EDTA; solvent: DMF	solvothermal syn-thesis	67	460	from 450 to 600	(0.32, 0.33) (white)	6565 (white)	68.4 (white)	32.26 (white)	Polyamideresin-650 (P-650)	red, orange, yellow, white, cyan, blue, and dark blue	white and multicolor N-doped CDs; color depends on molar ratio between the precursors	[109]
	ammonium citrate (in DMF), ethyl alcohol or ammonium citrate (in DMF), AEAPMS	solvothermal syn-thesis	51	395	~560	(0.33, 0.34)	6735	51	25.63	glass matrix	white	white CD-based glass thin films fabricated by screen printing technology	[160]
	glucose, ammonia	hydrothermal synthesis	10.2	~395	~590	(0.28, 0.37)	NR	NR	NR	PVA	white	synthesis of nitrogen-doped carbon dots (NCDs) and carbon sheets (NCSs); PVA used to prevent AIQ	[93]
	hexadecyltrimethyl ammonium bromide		38.7	~385	two broad bands at 495 and 660		3623–8121			PVP	white	modulation of the emitting states of colloidal CDs	[163]
2018	p-PD, ethanol, N-(3-(trimethoxysilyl) propyl) ethylenediamine (KH-792)	solvothermal syn-thesis	41.72	400	480–720	(0.44, 0.42)	2951	NR	NR	silica	warm white	embedding of CDs into a silica matrix overcomes the AIQ	[94]
	phenol derivative, EDA	hydrothermal method	24.4 in water; 53.3 in ethanol	360	broad band	(0.3316, 0.3373)	5538	93.3	NR	transparent epoxy JH-6800MA and JH-6800MB	yellow–green, white	WLEDs fabricated by mixing yellow–green N-doped CDs, blue CDs, and red emission (Sr, Ca) AlSiN3:Eu powders	[164]
	pyrogallic acid, DMF	solvothermal syn-thesis	8 (in KH-792); 16.82 (in DMF solution)	365	560	(0.38, 0.48) (at 3.5 V)	4503 (at 3.5 V)	NR	NR	KH-792	white	green CD emission due to large conjugated sp2-domain promoted by DMF; high-stability	[165]
	amino silane (red CDs), N-(2-aminoethyl)-3-aminopropyltrimethoxysilane (green CDs)	alkali-induced method	80 (red); 49 (green)					90.2	68.58		trichromatic warm white	optical properties of red-emitting CDs modulated by the alkali-induced surface electronic states; mix of red- and green-emitting CDs	[166]
	p-PD; solvent: isopropanol	solvothermal synthesis	20.6	450	broad band (peak at 580–600)	(0.335, 0.319) 0.5 nm-thick; (0.348, 0.324) 1.0 nm-thick; (0.371, 0.341) 1.5 nm-thick	5359 (0.5 nm) 4772 (1.0 nm) 3994 (1.5 nm)	81 (0.5 nm) 84 (1.0 nm) 85 (1.5 nm)	NR	PVA	white	combination of red CD solid film and yellow Ce:YAG PiG (0.5, 1.0, 1.5 nm film thickness)	[167]
	citric acid, urea; solvents: water, DMF, ethanol, NaOH	hydrothermal method	34 (blue); 19 (green); 47 (red)	365	442 (blue); 545 (green); 620 (red)	(0.38, 0.34)	3913	91	10.2	PVA	warm white	tunable emissions from blue, green, and red CDs (by changing the reaction solvent)	[95]
	Substituted derivatives from perylene (3,4,9,10-nitroperylene) (refluxed with HNO3)	solvothermal treatment in an alkaline solution	81 (green); 80 (red)	460	508 615			92.9	71.75	methyltriethoxysilane (MTES) and APTES	trichromatic white	photoluminescent CDs with green and red emission switching using perylene as the precursor	[168]
	1,2,4-triaminobenzene, polyethylene glycol 200 (PEG 200); solvents: ethyl acetate, ethanediamine, oleylamine, DMSO	solvothermal method	from 10.8 to 25	460	from 473 to 624	(0.4557, 0.3840) (YAG/red CDs: 0.50)	2514 (YAG/red CDs: 0.50)	89.6 (YAG/red CDs: 0.50)	NR	silica	warm white (red CDs mixed with Ce3+:YAG)	CDs can be well-tuned from 473 to 624 nm in different solvents	[108]
	potato starch, EDS	microwave-assisted hydrothermal method	2.46 (only starch; non-doped CDs); 5.71 (starch + EDS; N-doped CDs)	375	560 (non-doped CDs); 430, 560 (N-doped CDs)	(0.38, 0.45) non-doped CDs; (0.33, 0.35) N-doped CDs	4329 (non-doped CDs); 5437 (N-doped CDs)	NR	NR	starch	white	potato starch is used as a carbon source for CD and as an encapsulant	[147]
	citric acid, 5-amino-1,10- phenanthroline (Aphen)	one-pot hydrothermal method	67 (red); 29 (white)	400	triple emission bands: 430 (blue), 500 (green), 630 (red)	(0.33, 0.33)	NR	92	30.5	poly(2-hydroxyethyl methacrylate) (PHEMA)	pure white	multicolor emissive CDs with multiple core@shell structure	[111]
	citric acid, urea	hydrothermal method	53.82 (blue CDs); 36.18 (green CDs); 12.73 (red CDs)	365	445 (blue CDs); 510 (green CDs); 600 (red CDs); from 400 to 800 (white)	(0.33, 0.32)	5237	83	NR	transparent wood (lignin removed by oxalic acid and choline chloride), PAA	trichromatic pure white	green preparation of transparent wood as encapsulant; stability tested for 7 days	[149]
	p-PD, 3-isocyanatopropyltriethoxysilane (IPTS)	pyrolysis	NR	460	570 (dichromatic LED); 500, 605 (trichromatic LED); @50 mA	(0.397, 0.428) (dichromatic LED); (0.385, 0.345) (trichromatic LED); @50 mA	3949 (dichromatic LED); 4494 (trichromatic LED); @50 mA	70 (dichromatic LED); 85 (trichromatic LED); @50 mA	15.88 (dichromatic LED); 22.00 (trichromatic LED); @50 mA	PMMA (dichromatic LED); APTES-gel (trichromatic LED)	dichromatic and trichromatic white	a novel approach to achieve up/down-conversion photoluminescence of CDs based on polarity dependence	[112]
	o-phenylenediamine (o-PD); solvent: water	hydrothermal method	25–35	InGaN blue LED (*λ* NR)	broad band	(0.353,0.371) @100–150 mA	~5400 @100–150 mA	~78 @100–150 mA	~45 @100–150 mA	PVA, silica gel	white	yellow-emitting N-doped CDs	[169]
	citric acid, urea	microwave-assisted heating method	25 (red CDs); 36 (green CDs)	450	broad band (peaks at 532, 630)	(0.33, 0.33)	5610	92	12	PVP (for red CDs); starch (for green CDs)	pure white	enhanced red emissive CDs-based phosphors with high QY	[113]
	citric acid, piperazine; (different mass ratio: 1:0.5, 1:1, 1:2 *w/w*)	microwave-assisted heating method	NR	395	broad band centered at ~560	(0.25, 0.28); mass ratio: 1:1 *w*/*w*	13601	NR	NR	silicone	bluish white	the mass ratio of the precursors not only has a great influence on the CD sizes, but can also affect their luminescence properties	[114]
	o-PD, urea	one-pot microwave-assisted hydrothermal method	4.23	420	440–700 (peak: 563)	(0.30, 0.30)	7915	NR	NR	PVA	white	rapid synthesis of yellow fluorescent CDs	[170]
	starch, EDA	one-step hydrothermal method	9.65	365	two peaks at 420 and 555	(0.33, 0.37)	5462	NR	NR	starch, PVA	white	starch used as a carbon source for CDs	[148]
	phenylenediamine isomers (o-PD, m-PD, p-PD) formamide solution	microwave heating	in ethanol: 14 m-CDs; 45 o-CDs; 8 p-CDs; in water: 11 m-CDs; 38 o-CDs; 6 p-CDs	390 (m-CDs); 450 (o-CDs, or p-CDs)	470 m-CDs; 550 o-CDs; 600 p-CDs	(0.2678, 0.2945) m-CDs; (0.3613, 0.3851) o-CDs; (0.2423, 0.1283) p-CDs	10,967 m-CDs; 4589 o-CDs; 3247 p-CDs	83 m-CDs; 87 o-CDs; 81 p-CDs	18.3 m-CDs; 21.0 o-CDs; 17.7 p-CDs	starch, silicone	cool, neutral, and warm white	phenylenediamine isomers (oPD, mPD, and pPD) used as precursors for producing multicolor emissive CDs	[115]
	pyromellitic acid (PA), diethylenetriamine (DETA) and thiourea.	one-pot solvothermal method	16.7	InGaN blue LED (*λ* NR)	611	(0.57, 0.42) (orange LED)	1745 (orange LED)	56 (orange LED)	NR	PMMA chloroform solution	orange, white	solvothermal route for the synthesis of nitrogen and sulfur co-doped CDs; orange emissive CDs	[116]
2019	p-PD, amino acetic acid, ethanol, EDA	solvothermal method	24.7 (red CDs)	360	broad band (peaks at 400, 465, 600)	(0.33, 0.33)	5612	89	NR	PVP	pure white	synthesis of blue-, green-, and red-emitting CDs with high dispersity both in aqueous and organic solvent; WLED obtained by mixing the three types of CDs	[171]
	o-PD, starch	one-step hydrothermal method	66.9	455	broad band centered at around 600	(0.3429, 0.2817)	4613	83	30.54	silicone	daylight white	highly efficient solid-state yellow-emitting CDs phosphors	[127]
	diammonium hydrogen citrate, urea	pyrolysis	NR	450	broad band	(0.31, 0.36) (composite fibers); (0.33, 0.34) (mix red phosphor and CDs)	6000 (composite fibers)	90 (composite fibers)	63.5 (mix red phosphor and CDs)	PVP	white	WLED fabricated by a commercial red phosphor (Sr_2_Si_5_N_8_:Eu^2+^) and N-doped CDS embedded in PVP; fabrication of electrospun composite fibers	[172]
	citric acid, urea	one-step gaseous detonation method (within milliseconds)	11.2	365	broad band centered at 534	(0.31, 0.42)	6249	NR	NR	water solution dripped on an optical lens and dried	white	rapid CD preparation by a one-step gaseous detonation approach	[63]
	citric acid, urea; solvent: DMF	one-step solvothermal treatment	5.3 (blue CD); 12.4 (green CD); 8.9 (yellow CD); 6.9 (orange-red)	365	450 (blue); 550 (green); 575 (yellow); 610 (orange-red); 440, 540–590 flat band (white)	(0.18, 0.21) blue; (0.34, 0.54) green; (0.49, 0.46) yellow; (0.58, 0.38) orange-red; (0.32, 0.33) white	4820	82.7	NR	PVA	blue, green, yellow, orange–red, white	study of temperature on the evolution of CD surface states and on the emissive properties of CD-based LEDs	[117]
	pyromellitic acid, pentaethylenehexamine (PEHA); (solvent: DMF; dopant: manganese chloride tetrahydrate)	solvothermal method	28.5 (orange Mn-doped CDs); 83 (green CDs); 70 (blue CDs)	365	NR	(0.15, 0.19) blue; (0.25, 0.50) green; (0.55, 0.44) orange; (0.32, 0.31) white	6216	NR	NR	PVA	blue, green, orange, white	orange, green, and blue emissive CDs have synthesized; orange CDs doped with Mn to improve QY	[118]
	citric acid, tri(hydroxymethyl) amino methane hydrochloride (Tris-HMA)	one-step pyrolysis	15 (CD@PS); 23 (CD@PEGMA)	365	~440 (only blue CD@PS)	(0.32, 0.31) (mix of blue CD@PS, green 8-quinolinol, red CdSe/ZnS QDs)	NR	NR	NR	styrene, azobisisobutyronitrile (AIBN)	white	CDs prepared via mussel-inspired chemistry; CDs decorated by catechol-terminated hydrophilic poly(poly(ethylene glycol) methyl ether methacrylate (PPEGMA) and hydrophobic polystyrene (PS)	[128]
	KHP, NaN_3_, boric acid (BA); solvent: formaldehyde	one-step microwave-assisted pyrolysis	67.8	365	broad band centered at 432	(0.17, 0.14) (15% mass ratio of D-CDs and resin)	>100,000	37 (15% mass ratio of D-CDs and resin)	1.37 (15% mass ratio of D-CDs and resin)	epoxy silicone resin	bluish white	preparation of diamond-like carbon (sp^3^C) structure-doped carbon dots (D-CDs) powder for WLEDs	[173]
	citric acid, branched poly(ethylenimine) (b-PEI; molecular weight: 2000)	one-step hydrothermal method	26	450	565 (WLED); 590 (yellow LED)	(0.34, 0.34) white; (0.56, 0.43) yellow	4850 (white); 1849 (yellow)	70.5 (white)	8.9 (white)	NR	yellow, white	4 ns of CD luminescence lifetime enabling the fabrication of WLEDs and high-performance visible light communication system	[96]
	Tobias acid, o-PD; solvents: formamide (blue CDs), ethanol (yellow CDs), sulfuric acid (red CDs)	one-step solvothermal method	50.8 (red); 25.4 (yellow); 65.1 (blue); 50 (white)	365	410–460 flat band, 560	(0.31, 0.32)	6135	NR	NR	hydrogel	white	multicolor tunable highly luminescent CDs; N, S-CDs with red dual emission	[174]
	glucosamine, 3-[2-(2- aminoethylamino)ethylamino] propyl-trimethoxysilane (NQ-62); solvent: acetone	one-pot solvothermal treatment		460	broad band 380–780, centered at 600	(0.269, 0.184) (CD concentration: 50 g/L); (0.340, 0.255) (CD concentration: 100 g/L); (0.355, 0.268) (CD concentration: 120 g/L); (0.427, 0.327) (CD concentration: 200 g/L); (0.547, 0.383) (CD concentration: 250 g/L)	100,000 (CD concentration: 50 g/L); 4615 (CD concentration: 100 g/L); 3647 (CD concentration: 120 g/L); 2345 (CD concentration: 200 g/L); 1675 (CD concentration: 250 g/L)	42.3 (CD concentration: 50 g/L); 67.5 (CD concentration: 100 g/L); 68.1 (CD concentration: 120 g/L); 78.0 (CD concentration: 200 g/L); 81.9 (CD concentration: 250 g/L)	NR	no	white	synthesis of organosilane-functionalized carbon quantum dots (Si-CDs)	[119]
	poly(diallyldimethylammonium chloride) (PDDA)	microwave-assisted hydrothermal carbonization	~11.0 in water; 7.3 in acetonitrile; 8.1 in DMF; 7.3 in methanol; 1.4 in acetone; 1.3 in PVA	350	broad band	(0.303, 0.332) in PVA; (0.307, 0.354) in water	7023 in PVA; 5999 in water	NR	NR	PVA	white	CDs prepared starting from hydrothermal carbonization of PDDA; WLEDs fabricated by CD dispersion in water solution or in PVA	[175]
	citric acid, EDA	sonochemical synthesis	9–11	365	450−850	(0.334, 0.334) (optimized)	4290–6606	88–94	NR	lanthanoid metal–organic frameworks (Ln-MOFs)	white	fabrication of CDs/Ln-MOFs hybrids for WLEDs and as luminescent security inks	[90]
	poly(methyl methacrylate-co-dimethyl diallyl ammonium chloride) (PMMA-co-DMDAAC)	pyrolysis	NR	360	450	(0.1506, 0.0290) 1:5 molar ratios of DMDACC to MMA	NR	NR	11.24 (1:5 DMDACC:MMA)	patterned PMMA-co-DMDAAC composite	blue	honeycomb-patterned films of different pore sizes as a matrix increases luminous efficiency	[98]
	citric acid, urea; solvents: water, ethyl alcohol, DMF	hydrothermal method	NR	365	broad band (peaks located at 441, 536, and 622)	(0.3497, 0.3045) (white)	4878	85.2	NR	silica	blue, green, yellow, orange, red, white	multicolor emission CDs synthesized by varying the ratio of precursors, the solvents, the temperature, and reaction time	[121]
	p-PD, ZnCl_2_ solvent: ethanol	one-step solvothermal method	5.97 (pPD: ZnCl_2_ = 1:0—red); 6.24 (pPD: ZnCl_2_ = 1:0.1—purplish red); 6.92 (pPD: ZnCl_2_ = 1:0.5—purplish blue); 19.81 (pPD: ZnCl_2_ = 1:1—blue)	365	400–670 (two peaks located at 440, and 580)	(0.3301, 0.3367)	5606	89	NR	PVP	white	facile preparation of single metal-doped CDs with color-tunable properties	[120]
	phthalic acid and piperazine	microwave-assisted method	20.5 (in solid-state form and in aqueous solution)	450	520	(0.25, 0.32) CDs only; (0.33, 0.33) CDs + (Sr, Ca)AlSiN_3_:Eu	11,229 CDs only; 5618 CDs + (Sr, Ca)AlSiN_3_:Eu	62.5 CDs only; 88 CDs + (Sr, Ca)AlSiN_3_:Eu	87.7 CDs only; 64.4 CDs + (Sr, Ca)AlSiN_3_:Eu	silicone	bluish-white, pure white	fluorescent CDs in solid state form (using phthalic acid and piperazine as precursors)	[176]
	o-PD, dopamine	hydrothermal method	33.96	365	560–780 centered at 745 (red emission); broad band with three peaks at 400, 490, 600, 740 (multicolor white emission)	(0.68, 0.31) (red emission); (0.34, 0.37) (white emission)	1000 (red emission); 4957 (white emission)	71.8 (red emission); 81.9 (white emission)	NR	PMMA	red, white	preparation of red-emitting CDs, quenching in the presence of Fe^3+^; the CDs can be used as Fe^3+^ detectors in living cells	[177]
	citric acid, urea; solvent: DMF	pyrolysis	25.0	380	450−750	(0.35, 0.36) at 20 °C; (0.32, 0.23) at 80 °C	4075	93.2	14.8	polystyrene (PS)	warm and cool white	combination of blue and orange emissive CDs for WLEDs; temperature-dependent emission performance (emission spectrum turns from white (400−730 nm) at 20 °C to blue (∼440 nm) at 80 °C	[178]
	aluminum glycine (Al-Gly)	one-step decomposition route	15.6 *λ*_exc_: 400; 14.5 *λ*_exc_: 450	400, or 450, or 465	broad band with peak between 560 and 580	(0.3466, 0.3493) *λ*_exc_: 400; (0.3484, 0.3578 *λ*_exc_: 450; (0.3553, 0.3547) *λ*_exc_: 465	4935 *λ*_exc_: 400; 4898 *λ*_exc_: 450; 4637 *λ*_exc_: 465	66.9 *λ*_exc_: 400; 79.3 *λ*_exc_: 450; 80.6 *λ*_exc_: 465	1.0 *λ*_exc_: 400; 6.9 *λ*_exc_: 450; 10.4 *λ*_exc_: 465	AlOOH	white	preparation of CD-doped boehmite composite (CDs@AlOOH) alleviates the self-quenching effect	[179]
	urea	pyrolysis	25 (fluorescence); 6 (phosphorescence)	380	380–780 (peaks at 450, 570)	(0.35, 0.39)	4935	85	NR	cyanoacrylate (Super Glue)	white	blue–yellow fluorescence; phosphorescence dual emission	[180]
	resorcinol	solvothermal treatment	72 (pure red); 75 (pure green)	450 (FWHM: 18)	522 (FWHM: 32); 615 (FWHM: 33)	(0.35, 0.33) at 20 mA	NR	56.9 (at 20 mA)	86.5 (at 20 mA)	PMMA	pure red, pure green, white	wide color gamut CD-based LEDs for backlight displays (color gamut: 110% NTSC); high color purity narrow bandwidth emission (30 nm) triangular CDs	[12]
	citric acid, urea	microwave-assisted heating method	11	450	broad band (peak centered between 540 and 570)	(0.22, 0.23) volume ratio CDs@MMT: epoxy = 4:3; (0.36, 0.38) volume ratio CDs@MMT: epoxy = 5:3; (0.46, 0.49) volume ratio CDs@MMT: epoxy = 8:3	53131 volume ratio CDs@MMT: epoxy = 4:3; 4598 volume ratio CDs@MMT: epoxy = 5:3; 3232 volume ratio CDs@MMT: epoxy = 8:3	NR	NR	montmorillonite (MMT) clays, epoxy-silicone resin	bluish-white, yellowish	preparation of green emissive CDs@montmorillonite (CDs@MMT) composites	[181]
	citric acid, L-cysteine (CYS), KCl	one-pot microwave heating method	65	460	broad band (peak at 550)	(0.29, 0.38) 63.6% CD concentration; (0.32, 0.42) 70% CD concentration	NR	NR	97.8 (63.6% CD concentration); 93.9 (70% CD concentration)	PDMS	from cool to warm white	enhancement of yellow CDs fluorescence by AIE	[101]
	3,5-diaminobenzoic acid (DABA); 3,4-DABA; phosphoric acid—not blue CDs	one-pot solvothermal method	B-CDs, G-CDs and 29.2 (blue); 69.2 (green); 24.8 (red)	365	400–700	(0.2963, 0.3225),	7452	NR	NR	transparent silicone resin	cool white	preparation of high-emitting RGB-CDs with excitation-independent; fabrication of WLEDs by multicolor RGB-CDs	[182]
	citric acid, various hydroxyl-containing amino compounds	microwave-assisted heating method	from 56.9 to 87.0	405	472, 503, 527, 578, 629, 698	(0.332, 0.335) neutral WLED; (0.430, 0.405) warm WLED	5475.4 neutral WLED; 3058.7 warm WLED	96.6 neutral WLED; 96.4 warm WLED	46.8 neutral WLED; 41.2 warm WLED	epoxy resin	multicolor; warm and neutral white	preparation of tunable fluorescent CDs over the whole visible region; warm and neutral WLEDs are produced by coating cyan- and red-emitting CD layers on 405 nm LED chips	[13]
	expanded polystyrene (WEPS), dichloromethane (DCE), HNO_3_	one-step solvothermal method	5.2 (white solid-state CDs); 3.4 (yellow solid-state CDs); 3.1 (orange solid-state CDs)	365	broad band (peaks at 440, 550, 730)	(0.34, 0.39) white	5199 (white)	80.0 (white)	NR	PDMS	yellow, orange, red, warm-white,	preparation of CDs based on WEPS as the precursor to fabricate LEDs	[79]
2020	2,7- dihydroxynaphthalene (C_10_H_6_(OH)_2_); N, *N*-dimethylformamide (DMF, C_3_H_7_NO); solvent: ethanol	solvothermal method	26.03	360	480–680	(0.41, 0.39)	3330	91	NR	silane coupling agent (KH-792)	warm white	preparation of wide-spectrum orange-emitting CDs; KH-792 suppresses quenching	[97]
	citric acid, EDA	pyrolysis	56.6 (CDs/ PVP); 79.2 Ag/CDs/PVP	380	435	(0.156, 0.110)	NR	NR	39.2	PVP, PMMA photonic crystal; AgNO_3_ (for “islands” Ag film)	blue	blue CDs embedded in PVP and coupled with “island” Ag film result in a fluorescence enhancement	[139]
	citric acid, safranine T (ST)	one-step hydrothermal methodology	15.2 (liquid); 39.9 (solid)	365	599 (red); ~450, ~510, ~590,	(0.62, 0.36) red CDs; (0.33, 0.34) white CDs (with BMA:Eu^2+^ and BaSrSi:Eu^2+^)	5347 (white)	81 (white)	19.11 (white)	hardener (not specified)	red, white	red emissive host−guest CDs using citric acid (precursor and host) and ST (guest) as precursors; vitamin B12 quenches the fluorescence of CDs	[183]
	o-PD, NH_3_	hydrothermal method	NR	395	broad band with two peaks: 448 (zinc borate), 553 (CDs)	from (0.2366, 0.2550) to (0.4563, 0.5089)	NR	NR	NR	zinc borate matrix	blue, yellow, white	preparation of zinc borate/yellow N-doped CDs composite for WLEDs	[99]
	caramelized sugar (sucrose),	microwave-assisted heating method	~5	460 (blue LED), 520 (green LED)	broad yellow emission at 582 (blue exc.); broad red emission at ~630 (green exc.)	(0.31, 0.32) blue exc.; (0.4, 054) green exc.	NR	NR	NR	caramelized sugar	white, yellow	caramelized sugar CDs for color conversion applications	[150]
	L-tyrosine (for blue CDs), o-PD (for green CDs), L-tyrosine/o-PD mixture (for orange-red CDs)	hydrothermal method	8.6 (blue CDs); 12.6 (green CDs); 20.9 (orange-red CDs)	370	450 (blue); 545 (green); 580 (orange); 380−750 (white)	(0.23, 0.26) blue LED; (0.34, 0.43) green LED; (0.41, 0.42) orange LED; (0.30, 0.33) white LED	6293 (white)	83 (white)	NR	PVA	blue, green, orange, cool white	tunable emission colors in CDs by changing the molar ratio of suitable carbon sources	[146]
	1,2,4-triaminobenzene dihydrochloride, urea; (solvents: water, ethanol, tetrahydrofuran (THF), ethyl acetate (EtOAc), acetone)	solvothermal method	42 (blue CDs—acetone); 70 (green CDs—EtOAc); 79 (yellow CDs—THF); 64 (orange CDs—ethanol); 55 (red CDs—water)	UV (*λ* NR)	NR	(0.33, 0.45) white LED	5440 (white LED)	NR	NR	PVA	green, yellow, orange, white	multicolor emissive CDs based on solvent-controlled and solvent-responsive approaches	[122]
	ammonium citrate tribasic, formamide, glycerol, ethylene glycol	microwave-assisted method (at atmospheric pressure)	37.4	365	broad band (Ce:YAG/CDs @MPS)	(0.344, 0.333) (Ce:YAG/CDs@MPS)	4962 (Ce: YAG/CDs @MPS)	90.9 (Ce: YAG/ CDs @MPS)	67.5 (Ce: YAG/ CDs@ MPS)	mesoporous silica (MPS), epoxy AB glue	warm white	synthesis of red CDs; glycerol and formamide promote the carbonization precursor and enhance the crystallinity	[184]
	citric acid, EDA	electrostatic adsorption between positively charged QDs and negatively charged CDs	35	365	450–700	(0.32, 0.33) cool white; (0.37, 0.39) warm white	6338 (cool white); 4248 (warm white)	91 (warm white)	16.8 (warm white)	PMMA	cool and warm white	WLED prepared by core–shell structure nanocomposites based on Ag-In-S /ZnS@SiO_2_ QDs (AIS@SiO_2_) and CDs	[185]
	citric acid, urea; (solvent: water)	microwave-assisted heating method	62	450	broad band (peak at 540)	(0.29, 0.33)	7557	NR	42	epoxy silicone resin	cool white	a composite phosphor (CDs@g-C_3_N_4_) comprising carbon dots (CDs) and graphitic carbon nitride (g-C_3_N_4_) is prepared in water on a large scale	[186]
	maleic acid, m-phenylenediamine (m-PD)	room-temperature synthesis	42 (blue CDs); 35 (green CDs)	365	460 (blue LED); 520 (green LED)	(0.1993, 0.2423) blue LED; (0.3199, 0.5027) green LED; (at 80 mA)	NR	NR	16.6 (blue LED); 17.1 (green LED); (at 80 mA)	ethanol dropped on an optical lens	blue, green	no external energy or irradiations, reactants or high temperature are required to prepare CDs	[187]
	phosphoric acid, urea	one-step process involving polymerization, deamination, and dehydration reactions	~41 (white fluorescence) 23 (green phosphorescence)	370	380–580	(0.268, 0.346)	8756	85.3	18.7 (at 20 mA)	silicone	cool white	carbonized polymer dots (CPDs) white light with dual components consisting of simultaneous fluorescence (S_1_→S_0_) and phosphorescence (T_1_→S_0_)	[125]
	citric acid; 1-(2-pyridylazo)-2-naphthol (PAN); solvents: DMSO, DMF, ethanol, (THF), dichloromethane (DCM)	one-step solvothermal method	46.5 (purple CDs); 32.3 (blue CDs); 31.6 (red CDs)	365	375, 455, 570	(0.29, 0.31) in epoxy resin	NR	NR	NR	epoxy resin; PMMA	purple, blue, red, white	multicolor emission CDs synthesized by one-step solvothermal treatment using citric acid and PAN with concentration-tunable fluorescence and solvent-affected aggregation states	[188]
	citric acid, nitric acid, 1-octadecene, oleylamine (OLA), methanol, nitrogen	microemulsion process	NR	360	450, 500, 590, 690 (D65)	(0.35, 0.37) D50; (0.32, 0.33) D65	6354 (D65; at 120 mA)	95.3 (max)	17.68 (D65; at 20 mA); 7.93 (D65; at 120 mA)	PMMA/toluene solution	D50, D65 white	synthesis of a sun-like light source (D50, D65) with tri-chromatic broad spectra (435-nm CDs), 695-nm perovskite QDs, and dual peak 510- and 590-nm Ag-doped InP QDs	[189]
	citric acid, EDA (blue CDs); citric acid, urea (green CDs)	pyrolysis (blue CDs); microwave-assisted heating method (green CDs)	>60 (blue, green)	390	450–700	(0.3514, 0.3715) CD@SiO_2_ + red CdTe@CaCO_3_	4850 (CD@ SiO_2_ + red CdTe@ CaCO_3_)	89.1 (CD@SiO_2_ + red CdTe@ CaCO_3_)	NR	spherical SiO_2_ matrix, cetyltrimethylammonium bromide (CTAB)	blue, green, red (CdTe@ CaCO3), white	blue/green-emitting N-doped carbon dots embedded into silica nanospheres (CD@SiO_2_) with spherical morphology	[190]
2021	potassium bisulfate, acetic acid, hydrochloric acid; m-PD	solvothermal method	NR	365 (for green and yellow LEDs); 460 (for white LEDs)	500 (green); 550 (yellow); 560 (white)	(0.29, 0.49) green; (0.39, 0.49) yellow; (0.26, 0.27) white	NR	NR	NR	PVA	green, yellow, white	acid catalyst induces a fluorescence red shift and improved the QY of green emissive CDs.	[191]
	citric acid, EDA	hydrothermal method	4.57 (blue CDs only) 40.1 (CS composites)	365	445	(0.15, 0.13) at 35 mA	NR	79.1	NR	silicone	deep blue	synthesis of CD-silica (SiO_2_) spheres composites (CS composites) with 10-fold fluorescence enhancement	[141]
	1,4-diaminonaphthalene (solvents: octane, toluene, ethanol, acetone, DMF)	solvothermal method	26.4 (in octane); 17.9 (in acetone); 13.9 (in toluene); 12.1 (in ethanol); 7.5 (in DMF)	365	427–679 (peak at 618)	(0.4175, 0.2936)	NR	NR	NR	epoxy resin	red	preparation of red CDs using different solvents for carbonization; CDs prepared in octane have the high QY and fluorescence intensity	[192]
	citric acid, urea (solvent: DMF)	solvothermal method	22.7 (exc. 340); 28.1 (exc. 360); 24.0 (exc. 380); 23.8 (exc. 400); 15.6 (exc. 420); 14.2 (exc. 440); 10.0 (exc. 460); 9.8 (exc. 480)	460	broad band (up to 600) centered at 540 (CDs only); two large peaks at 470 and 535 (CDs + Ce:YAG)	(0.33, 0,45) CDs + Ce:YAG (CCT: 5602)	5602–9242	NR	NR	PVA (MW: 1500)	cool and warm white	WLEDs fabricated with an adjustable CCT by combining Ce:YAG and green-emitting CD films	[193]
	dehydroabietic acid, ethanolamine	one-pot hydrothermal reaction	10	450	500–675	(0.3304, 0.3055)	5608	88.6.	NR	epoxy resin	white	steric hindrance is exploited to prepare biomass-based solid-state emissive CDs with high yield	[151]
	1,6-dihydroxynaphthalene (1,6-DHN); L-asparagine (L-Asn)	solvent-free carbonization method	NR	365	451, 518	(0.32, 0.31)	NR	NR	NR	polyvinyl butyral (PVB)	pure white	white light-emitting CDs prepared through a solvent-free method	[194]
	polyethyleneimine (PEI), phosphoric acid, ethanol	solvothermal method	4.4 (in aqueous solution or in powder form)	365	broad band	(0.33, 0.33)	3319.6	76.3	NR	AB glue	pure white	novel kind of self-quenching-resistant N,P-doped CDs; blue-emitting in solid state form and mixed with a red emitting Eu-MOF powder (CD@Eu-MOF)	[195]
	o-PD, ammonia water (NH_3_·H_2_O, 25 wt%),	hydrothermal synthesis method	NR	365	broad band	from (0.2798, 0.2916) to (0.3246, 0.3305) (ZBH:9%Tm^3+^/yNCDs)	NR	NR	NR	ZBH:9% Tm^3+^	white	incorporation of yellow N-doped CDs in a Tm^3+^-doped zinc borate (4ZnO·B_2_O_3_· H_2_O, ZBH) with a flake-like morphology	[196]
	*Pithecellobium dulce* leaves (solvent: DMF)	muffle furnace carbonization method	NR	365	530	(0.32, 0.43) PVDF/CDs film dried at room temperature (white); (0.33, 0.42) PVDF/CDs film dried at 65 °C (greenish-white)	5863 (white); 5576 (greenish-white)	NR	NR	polyvinylidene fluoride (PVDF)	white, greenish white	films doped with plant derived photoluminescent CDs	[197]
	EDA, trimethylolpropane tri(cyclic carbonate)ether (TPTE) (synthesized from the reaction of CO_2_ and trimethylolpropane triglycidyl ether in ethanol)	solvothermal treatment	46.2 (in solution) 11.3 (as a solid)	365 (orange); 420 (white, yellow); 460 (red)	620 (orange); 605 (yellow); 675 (red); ~610 (white)	(0.582, 0.413) orange; 0.549, 0.443) yellow; (0.545, 0.309) red; (0.376, 0.288) white	3161 (white)	85 (white)	NR	concentrated CPDs optical lens (no encapsulant)	orange, yellow, red, warm white	new type of carbon dioxide (CO_2_) derived CPD; SSF due to the self-passivation of poly(hydroxyurethane) chains on the surface of the carbon core	[198]
	citric acid, urea	microwave-assisted heating method (700 W, 5 min)	5.78	460	broad band	(0.42, 0.51)	NR	NR	NR	zeolitic imidazolate framework 8 (ZIF-8), epoxy resin	white	green-emitting CDs and red-emitting rhodamine B (RhB) molecules, encapsulated into porous ZIF-8 to obtain a yellow-emitting composite phosphor	[199]
	avocado peel (CPDs-P); sarcocarp (CPDs-S)	hydrothermal method	9.56 (CPDs-P); 8.97 (CPDs-S)	365	broad band centered at 600 (warm white); broad band centered at 485 (cool white)	(0.38, 0.39) warm white; (0.29, 0.34) cool white	4088 (cool white)	90.47 (warm white); 84.54 (cool white)	NR	epoxy resin	warm and cool white	blue-emitting CPDs-P (peel) and blue–green-emitting CPDs-S (sarcocarp) prepared using the peel and sarcocarp of avocado, respectively	[152]
	2-aminoterephthalic acid (ATA), polyethylene glycol, orthophosphoric acid (H_3_PO_4_).	microwave-assisted pyrolysis	67	365	broad band centered at 578	(0.35, 0.33)	5246	92	NR	PVA	pure white	synthesis of multifunctional CDs for WLEDs, ultrasensitive to the nitroaromatic explosive picric acid	[200]
	citric acid, octadecene, hexadecyl amine (HDA) (solvent: chloroform)	open-air atmosphere carbonization synthesis	5–13 (in colloidal state); 16 (in solid state)	350	360–700 (peak: 430, FWHM: ~154)	(0.31, 0.33)	6412	~ 96	NR	PDMS	pure white	ecofriendly open-air atmosphere synthesis of highly luminescent CPDs for high CRI WLEDs	[129]
	ammonium citrate (red CDs); AEAPMS (+red CDs) (white CDs); solvent: DMF	pyrolysis	NR	395	broad band (peaks: 540, 645)	(0.357, 0.359)	4271	89	1.3	cellulose acetate	white	novel solvent engineering by controlling the dilution ratios between the solvent (DMF) and pristine red CDs solution	[201]
	citric acid; 1,4,7,10-tetraazacyclododecane (cyclen)	microwave-assisted synthesis method	48	450	545 (CDs only); 545, 595 (CDs + (Sr, Ca) AlSiN_3_: Eu)	(0.30, 0.34) (CDs only); (0.33, 0.32) (CDs + (Sr, Ca) AlSiN_3_:Eu)	7001 (CDs only); 5557 (CDs + (Sr, Ca) AlSiN_3_: Eu)	57.7 (CDs only); 82.1 (CDs + (Sr, Ca) AlSiN_3_: Eu)	25.5 (CDs only); 31.0 (CDs + (Sr, Ca) AlSiN_3_: Eu)	silicone	white	bright yellow fluorescence from cyclen-based CDs powder	[81]
	citric acid, urea, oleic acid	microwave-assisted solvothermal reaction	0.99 (CDs embedded in PVA)	420	broad band (peak: 510–530)	(0.39, 0.46)	4105	NR	NR	PVA	white	PVA polymer encapsulated with N-doped CDs sustaining the emission in its solid state	[202]
	spinach after soaking in ethanol/water (4:1 vol ratio)	one-step hydrothermal method	NR	395	447, 677	(0.185, 0.104) blue; (0.352, 0.339) white; (0.433, 0.375) orange	NR	65–82	12.4–19.5	Mg(OH)_2_ nanosheets, ethylene vinyl acetate copolymer (EVA)	blue, white, orange	blue/red CDs@Mg(OH)_2_ anti-self-quenching luminescent composites for plant growth applications	[153]
	maleic acid, APTES	one-step solvothermal method	34.06 (green CDs); 38.07 (yellow CDs); 20.3 (orange CDs)	365	475 (blue LED); 460, 510 (yellow LED); 560 (orange LED)	(0.22, 0.41) green LED; (0.38, 0.47) yellow LED; (0.50, 0.43) orange LED	9885 (blue LED); 6772 (yellow LED); 2379 (orange LED)	70.2 (blue LED); 72.6 (yellow LED); 74.5 (orange LED)	NR	epoxy resin	blue, yellow, orange	SSF Si-doped CDs prepared using maleic acid and APTES as precursors	[203]
	citric acid, thiourea, ammonium fluoride (solvent: DMF)	solvothermal method	22.64	525 (for red LEDs); UV (*λ* NR) (for white LEDs)	695 (red LEDs); 425, 530, 650 (white LEDs)	(0.72, 0.28) red LEDs; (0.36, 0.36) white LEDs	1000 (red LEDs); 4705 (white LEDs)	57.7 (red LEDs); 93.8 (white LEDs)	NR	epoxy resin	red, white	preparation of highly stable near-IR CDs (emission at 714 nm) using citric acid as the carbon source, thiourea and ammonium fluoride as the dopant source	[123]
	2-amino-1-naphthol, EDA (solvent: ethanol)	one-pot hydrothermal method	NR	452	550 (yellow–green CDs), 628 (red Cd^2+^-based QDs)	(0.3669, 0.3671)	4329	95.1	NR	BaSO_4_, silica gel	white	improvement of yellow–green CDs by encapsulation in BaSO_4_	[204]
	cis-butenedioic acid (C-BA), urea	one-step solvothermal method	35.12	460	broad band centered at 580	(0.3341, 0.3075)	5388	86.9	15.12	Ca(OH)_2_, transparent silicone (YD65-5A), anhydride curing agent (YD65-5B)	pure white	multicolor CDs obtained from cis-butenedioic acid (C-BA) and urea; solid-state luminescent CD obtained by adding alkaline Ca(OH)_2_	[205]
	citric acid, EDA, Ln(NO_3_)_3_∙6H_2_O	solvothermal and hydrothermal methods	NR	275	narrow bands: ~420, ~475, ~530, ~580, ~610, ~680	(0.337.0.339) neutral white	5319 (neutral white)	93 (neutral white)	322 (warm white)	NR	cool, neutral, warm white	rare-earth single-atom-based NaGdF_4_:Tb^3+^/ Eu^3+^@CDs:N/ Eu^3+^ composite with tunable full-color luminescence	[206]
	citric acid; (3-aminopropyl)triethoxysilane; N-[3-(trimethoxysilyl)propyl]ethylenediamine (silanes molar ratio = 7:3)	one-step solvothermal method	~20	365, 450	550 (exc. 365, yellow emission); 575 (exc. 450, white emission)	(0.41, 0.52) yellow; (0.33, 0.31) white	5774 (white)	81.6 (white)	NR	no encapsulation	yellow, white	double silane-functionalized carbon dots (DSi-CDs) with emission at longer wavelengths	[207]
	citric acid, tris(hydroxymethyl)methyl aminomethane	microwave-mediated heating; liquid-liquid diffusion-assisted crystallization method	92.7 (CDs only)	380	430, 600	(0.3580, 0.3611)	4255	93.6	12.64	NR	white	assembly of blue-emitting CDs and yellow-emitting Cs_2_InCl_5_·H_2_O: Sb^3+^ metal halide crystals for WLEDs (CDs@Cs_2_InCl_5_·H_2_O: Sb^3+^)	[208]
	gallic acid, o-PD (solvents: methanol, ethanol, acetone, octane, toluene, THF, DMF)	solvothermal method	10 (exc. 340, blue CDs); 11 (exc. 500, green CDs); 23 (exc. 500, red CDs)	purple (*λ* NR)	425 (blue), 510 (green), 585 (red); broad band (white)	(0.16, 0.12) blue; (0.27, 0.45) green; (0.53, 0.38) red; (0.32, 0.34) white	NR	NR	NR	epoxy resin	blue, green, red, white	blue, green, and red CDs prepared using gallic acid as the raw material	[209]
	tartaric acid, triammonium citrate	one-step solvothermal method	43.6 (blue CDs); 41.2 (green CDs); 44.1 (red CDs)	365	465, 530, 570	(0.17, 0.15) blue; (0.31, 0.53) green; (0.58, 0.40) red; (0.34, 0.35) white	5336 (white)	83.1 (white)	NR	KH-792 silane coupling agent	multicolors, warm white	synthesis of multicolor fluorescent CDs with adjustable emission wavelength and high QY, using ecofriendly precursors	[210]
	2,3-diaminopyridine (solvents: NaOH, pure water, HCl)	pH-controlled hydrothermal method	8.4 (violet); 9.3 (green); 8.3 (orange)	365	~400, ~540, ~600	(0.22, 0.17) bluish; (0.30, 0.33) cool white; (0.41, 0.37) warm white	7466 (cool white); 3265 (warm white)	78 (cool white); 90 (warm white)	NR	starch, AB glue	bluish, cool white, warm white	single precursor to synthesize colorful CDs in different pH conditions	[211]
	citric acid, urea	microwave-assisted pyrolysis	15 (exc. 395); 14 (exc. 410)	400	~500, ~610	(0.321, 0.367) cool white; (0.390, 0.455) warm white	5796 (cool white); 4228 (warm white)	89 (cool white); 84 (warm white)	13.2	d-U(600) di-ureasil hybrid matrix	cool and warm white	assembly of a commercial LED and flexible films of a di-ureasil hybrid comprising a mixture of a cyan component originating from CDs and a red one resulting from the Eu(tta)_3_(bpyO_2_) complex	[212]
	citric acid, urea, thiourea	microwave-assisted solvent-free synthesis method	NR	410, 455	605	NR	NR	NR	NR	NR	NR	synthesis of graphitized N-doped and S,N-doped CDs	[213]
	Star Jasmine leaves (*Trachelospermum jasminoides*)	solvothermal treatment	54.8 (blue CDs); 15 (green CDs); 19.8 (red CDs)	NR	466 (blue); 521 (green); 625 (red)	(0.15, 0.18) blue; (0.26, 0.60) green; (0.58, 0.31) red; (0.36, 0.32) white	4283 (white)	NR	NR	polyurethane	blue, green, red, white	synthesis of RGB CDs with high optical tuning using sustainable green precursors	[214]
2022	o-PD, phenylalanine	one-pot solvothermal method	52.34 (blue CDs); 65.67 (green CDs); 12.87 (red CDs)	365	400–750 (peaks: 400, 500)	(0.30, 0.35)	7127	86	NR	epoxy resin, tetraethylenepentamine	pure white	a RGB-multicolor CDs prepared by adjusting the type of reaction solvent	[215]
	EDA, nitrogen	hydrothermal method	NR	395	broad band (peaks: 450, 475, 580, 610, 700)	(0.18, 0.16) (KLM: Eu^3+^: CDs = 0:1, blue LED); (0.19, 0.21) (KLM: Eu^3+^: CDs = 1:2, cyan LED); (0.21, 0.22) (KLM: Eu^3+^: CDs = 1:1, cool white LED); (0.29, 0.25) (KLM: Eu^3+^: CDs = 2:1, neutral white LED); (0.38, 0.28) (KLM: Eu^3+^: CDs = 10:1, warm white LED); (0.56, 0.32) (KLM: Eu^3+^: CDs = 1:0, red LED);	NR	NR	NR	silica sol	blue, cyan, cool, neutral, and warm white, red	fabrication of polychromatic nanoplatform KLa(MoO_4_)_2_: Eu^3+^@CDs (KLM: Eu^3+^ @CDs) by encapsulating CDs on the surface of KLM: Eu^3+^ with silicon shell	[216]
	citric acid, Nile blue A (NBA)	one-pot solvothermal method	64 (blue CDs); 57 (yellow CDs); 51 (red CDs)	395	475, ~550, 625	(0.15, 0.21) blue LED; (0.43, 0.54) yellow LED; (0.46, 0.16) red LED; (0.31, 0.29) white LED	5643 (white)	87.2 (white)	NR	epoxy resin, curing agent EDA,	blue, yellow, red, white	synthesis of tunable multicolor emission CDs, covering the entire visible spectrum	[217]
	ethanol, H_2_SO_4_	one-step carbonization process	14.88 (blue CDs); 4.85 (cyan CDs); 17.54 (yellow CDs)	UV (*λ* NR)	~450, ~475, ~590,	(0.17, 0.21) blue LED; (0.24, 0.33) cyan LED; (0.50, 0.47) yellow LED; (0.37, 0.39) white LED	~4500	87.8 (white)	NR	epoxy resin	blue, cyan, yellow, white	white light-emitting CDs using ethanol as the carbon source and H_2_SO_4_ as the carbonizing agent with three emission centers (blue, cyan, yellow)	[218]
	phloroglucinol, urea	one-step microwave method	48.2 (blue CDs); 26.0 (green CDs); 18.5 (yellow CDs); 13.7 (orange); 5.7 (red)	UV (*λ* NR)	~450 (blue); ~520 (green) ~580 (yellow); ~600 (orange); ~670 (red); 430–650 (white)	(0.20, 0.18) blue LED; (0.31, 0.44) green LED; (0.48, 0.50) yellow LED; (0.52, 0.45) orange LED; (0.68, 0.31) red LED; (0.29, 0.31) cool white LED; (0.32, 0.36) pure white LED; (0.39, 0.40) warm white LED	8588 (cool white LED); 6104 (pure white LED); 3937 (warm white LED)	88 (cool white LED); 82 (pure white LED); 80 (warm white LED)	NR	optical sealant OE6250 A, and B on optical lens	blue, green, yellow, orange, red, cool, pure, and warm white	self-quenching-resistant solid-state fluorescent CDs displaying tunable full-color emission	[219]
	2-amino-1-naphthol; EDA (solvent: ethanol)	pyrolysis	NR	460	~545 ~625	(0.332, 0.336)	5498	93.9	32.19 (before ageing)	starch, epoxy resin	white	CD synthesized by adsorbing yellow–green CDs to starch particles and mixed with CdZnSeS/ZnS phosphor	[220]
	Polyethylene glycol (Mw = 4000) (PEG-4K); 1,2-diaminobenzene	hydrothermal method	62.5	455	400–700 (peak at 541)	(0.3003, 0.3731)	5117	80.9	40.6	mesoporous silica nanosphere-stellate (MSNS), silicone resin	pure white	CD@monodisperse mesoporous silica nanosphere-stellate (CD@MSNS) hybrid phosphor with highly concentrated emitting centers	[221]

## Data Availability

Not applicable.

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
