# Peer review of "Color Conversion Light-Emitting Diodes Based on Carbon Dots: A Review"

_materials, 2022, doi:10.3390/ma15155450_

Round 1
Reviewer 1 Report
The presented review discusses the preparation, properties and use of carbon dots in the preparation of light-emitting diodes. I have thoroughly read the review and I would like to state that it is very nicely done. I appreciate the work with an overview of articles that are summarized in Table 1. Especially since the authors made a sober inventory of the photoluminescence quantum yields of CDs, as this is a very questionable topic, especially if some scientists in some of their articles give only the highest values and keep the lowest ones silent.
My evaluation of the submitted review is positive, but I have a reservation about the quality of the images. It seems very low to me, so I would recommend authors to insert images with a higher resolution, if possible.
Another recommendation is to use a reference (https://doi.org/10.1039/D2RA01911F ) in which other possible uses of CDs that the authors did not mention are mentioned and demonstrated. This could also help broaden the readers' horizons. The reference would be suitable on p1 line33.
Reviewer 2 Report
This is a comprehensive, nice, easy-to-read and detailed review on carbon dots based LEDs. I recommend publication with minor revision.
Comments
Figure 1. Which source was used for calculation of the number of articles (WOS, Scopus, SciFinder…?)
Figure 2. This is a very classical Jablonski diagram: it is better to draw yourself or to take from more standard publication (book, review) rather than from a PhD thesis, and accordingly to show radiative and non-radiative transitions (solid and dotted/waved, respectively).
As I could notice, photoinduced electron transfer (PET) is discussed only related to quenching of fluorescence (p.6). However PET may lead also to broadening of the emitted luminescence. It should also be discussed.
